# The Carving of Kṛṣṇa's Legend: North and South, Back and Forth

**Charlotte Schmid**

French School of the Far East (École Française d'Extrême-Orient), 75116 Paris, France; charlotte.schmid@efeo.net

**Abstract:** This paper emphasizes the role played by the sculptural tradition in the elaboration of religious narratives that today are mostly studied through texts. It aims to demonstrate that according to the documents we know, the legend of Kṛṣṇa has been built through one continuous dialogue between different media, namely texts and carvings, and different linguistic areas, Indo-Aryan and Dravidian. Taking the motif of the butter theft as a basis, we stress the role played by the sculptural tradition and Tamil poetry, two elements less studied than others, at the foundation of a pan-Indian Kṛṣṇa-oriented heritage. We posit that the iconographic formula of the cowherds' station as the significant background of the infancy of Kṛṣṇa led to the motif of the young god stealing butter in the texts, through the isolation of one significant element of the early sculpted images. The survey of the available documents leads to the conclusion that, in the southern part of the peninsula, patterns according to which stone carvings were done have been a source of inspiration in Tamil literature. Poets writing in Tamil authors knew texts transmitted in Sanskrit, Prākrit, and Pāli, and they certainly had listened to some others to which we have no access today. But we give reasons to assume that the authors of the said texts were also aware of the traditional ways of representing a child Kṛṣṇa in the visual domain. With these various traditions, poets of the Tamil country in the later stage of Tamil Caṅkam literature featured a character they may not have consciously created, as he was already existent in the visual tradition and nurtured by the importance of one landscape animated by cowherds in the legend of Kṛṣṇa.

**Keywords:** Tamil Caṅkam; visual tradition; Kṛṣṇa's legend; Bhāgavata-purāṇa; Harivaṃśa; Divyaprabandham; textual transmission; Pallava; Cōḻa

## 1. Introduction

> "He loves butter
> How radiant!—fresh butter in his hand
> ( . . . )
> Blessed, says Sur, is one instant of his joy.
> Why live a hundred eons more?"
> Transl. John S. Hawley, *The Memory of Love, Sūrdās Sings to Krishna*. (Hawley 2009)

Thus begins and ends one of the many poems of the Sūrsāgar, an anthology whose literary elements started to be composed during the second half of the 16th century in the language of Braj (Brajbhāṣā), in the region of the same name whose center, Mathurā, is located around 100 km south of Delhi in northern India. The childhood of Kṛṣṇa, a deity born and raised in the area of Mathurā according to all the texts we know, and his love for Rādhā constitute the main focus of the Sūrsāgar's poems. The lovely child who loves butter, the mākhan-cor, "the Butter-thief," is the hero of an important section of the text. Since the struggle for the recognition of Hindi as a literary language in the 1920s and 1930s, during which the Sūrsāgar was used as a key tool, the Butter-thief assumed in the north of

India a singular importance, as has been amply demonstrated by John S. Hawley in a book published in 1983, *Krishna the Butter-thief* (Hawley [1983] 1989).

Given such a modern and contemporary background, the fact that the theft of butter by Kṛṣṇa is first mentioned in Tamil poetical works from the south of India may come as a surprise. This is all the more so, since, from the period of its composition between the ninth and 11th centuries, the well-known Bhāgavata-purāṇa has been cited as the main textual source for the deeds of the young god, inspiring paintings, carvings, and texts in which Kṛṣṇa is the hero. Composed in Sanskrit, the language of an elite koine active far beyond India and presenting a continuous and elaborated narrative of Kṛṣṇa's life, the Bhāgavata-purāṇa may have inspired much of the Sūrsāgar. However, Bhāgavata-purāṇa is far from being the first version of the story of the childhood of Kṛṣṇa. It was already told in Sanskrit in the much earlier Harivaṃśa, an appendix to the Mahābhārata dated between the second and the fourth centuries CE. Furthermore, the Bhāgavata-purāṇa was composed in a South Indian milieu, which raises another issue. Its author(s) were inspired by Tamil versions of Kṛṣṇa's legend that were being developed since early Tamil poetic anthologies and epics from the seventh century. The theft of butter does occur in those early works, but only rarely. In the Divyaprabandham (Nālāyira Tivviyappirapantam), a collection of Vaiṣṇava devotional poems composed between the seventh and the ninth centuries however, a butter lover variously called Kaṇṇaṉ (Kṛṣṇa) or Kēcavaṉ (Keśava), etc., plays an active role. As is true for several elements of Kṛṣṇa's narrative, the earliest Tamil poems surface as models for the stanzas of the Divyaprabandham, that, in turn, have deeply inspired the Bhāgavata-purāṇa when the latter introduced many "new" patterns into the Sanskrit corpus.

As in the case of many ubiquitous motifs of the legend of Kṛṣṇa, the role played by Tamil texts in the development of the Butter-thief figure is largely neglected, while their first inspiration remains a mystery. What was the access of the Tamil area to the legend of Kṛṣṇa? Were there versions of the Harivaṃśa or other texts, including oral versions that can only be hypothesized and were lost in the course of time? Jain texts have transmitted accounts of Kṛṣṇa's legend that do not correspond in all details with the one recorded in the Harivaṃśa. Are they one of the channels through which composers of the Kṛṣṇa story in Tamil got access to Kṛṣṇa-flavored stories? Or, at least, do they give witness to one of the links with earlier texts that are now lost? How is it that so many episodes of the legend of Kṛṣṇa made their appearance in the Tamil South and not in Sanskrit or another Indo-Aryan language? The Butter-thief is, in fact, one of the many motifs that first emerged in Tamil literature. Kṛṣṇa as a flautist, Kṛṣṇa stealing the clothes of the gopīs, Kṛṣṇa defeating a calf-demon or a heron are other instances of the creativity of Tamil poetry in the Kṛṣṇa-devoted realm.

But Tamil texts are not the only evidence neglected with regard to Kṛṣṇa's Bhakti stories. The role the visual tradition played in the development of the legend of Kṛṣṇa has still to be fully recognized. This essay proposes that art—only stone sculptures survive, but paintings or other movable media like terracottas or ivories may have existed—should be seen as the missing link between these two text-worlds: Northern and southern.

For the south of the peninsula, the patterns designed for depicting Kṛṣṇa on carvings provided one source of inspiration that was as important as were texts to southern poets. Certain motifs found in the early, North Indian carvings do not have clear parallels in the early northern texts, while later South Indian poems focus on them in unambiguous detail. This paper argues that the depiction of the cowherd village on the earliest stone carvings was one such motif; it would become the basis for inventing the trope of the theft of butter in poems composed in the South. The thief of butter was, in addition, connected in these early poetic anthologies with lovers as thieves of another kind. Through this complex interplay of sculpture and text emerged a new and irresistible portrait of Kṛṣṇa that was destined to have a remarkably long life in both northern and southern circles.

## 2. The Butter-Thief as a Northern Motif

Having sold butter, at the time of the return—does one remember?—On that day,
When the flowers of jasmine densely flourished, near the small river of a wood,

With your eyes similar to tender young mangoes, my heart
You took to bind (me) and rule on the battlefield: what if you are not but a thief of a
unique kind?
Kalittokai 108.26–29.[1]

The Sūrsāgar seems to anticipate representations of Kṛṣṇa familiar today. An example is the poem
that frames this paper, given in the accurate translation of J.S. Hawley (Hawley 2009, p. 52). Here is
the thief of butter:

( … )
Crawling on his knees, his body adorned with dust,
Face all smeared with curd,
Cheeks so winsome, eyes so supple,
A cow-powder mark on his head,
Curls swinging to and fro like swarms of bees
Besotted by drinking drafts of honey,
At his neck an infant's necklace, and on his lovely chest
The glint of a diamond and a tiger-nail amulet.
Blessed, says Sur, is one instant of his joy …

In many other verses of the Sūrsāgar, the deity takes butter, eats it, or is featured stealing or having
stolen it.[2] This thief is still the hero of scenes played in the area of Mathurā, of popular images, songs,
and videos in the whole of India and abroad, following that tradition that must have lain behind the
composition of the Sūrsāgar.[3]

The visual aspect of Kṛṣṇa's occurrences in the Sūrsāgar is striking. If songs, poems, plays, and
videos are related to oral modes of transmission, performances must also be included among the
possible sources of visual inspiration for many of these works, and poetry featuring the Butter-thief is
no exception. These kinds of vignettes are the visions of a poet. As in the poem just cited, the figure is
sharply delineated; colors are suggested (curd, powdered cow dung, bees),[4] as well as the radiance of
the skin (radiant, dust, cow dung-powder, swarms of bees, glint of diamond); signs are given that
indicate a young age (smeared with curd; winsome; cow dung mark; infant's necklace; tiger amulet).
The poet makes us see a god he describes in all his details—position, hairdo, ornaments, actions, things
held by the deity—as the source of wonder for him. In doing so, he privileges the sense of sight over
others.[5] The poet does not touch, smell, or taste this deity, nor does he listen to him.

To privilege sight in a corpus to be performed may seem too obvious for those who study Bhakti
texts, as natural as it may be for the devotees themselves. However, in the Tamil Bhakti corpus, smell
is of notable importance, and sound—above all the music of the flute played by Kṛṣṇa—is another
privileged sense. With regard to the episode of the butter theft, however, it is the visual components
that are of distinct importance, from the first texts that were composed in the South, to the northern
variations in the *Sūrsāgar* and other texts.[6] For the moment, let us stress that the Butter-thief vignette
is one of those that are clearly delineated in texts and that its characteristics and details are closely

---

[1]  Translations are my own unless otherwise noted.
[2]  For other references, see the two chapters "Sūr's Butter-thief poems, two types" and "the Butter-thief in context."
     (Hawley [1983] 1989, pp. 129–61).
[3]  For more (including bibliography) see the chapter titled "The Butter Thief Līlā" in (Hawley [1983] 1989, pp. 181–222).
[4]  Red, black, and white dominate the vignettes of the *Sūrsagar*. They may correspond to illustrations drawn with these
     three colors.
[5]  Seeing remains a master theme of the anthology, see *Sūrsāgar* 369, " … Tell him what I've described/And bring before these
     eyes, says Sūr, the image—." transl. Hawley (1987, p. 169). Also "Lost, lost, lost to Mohan's captivating image," from poem
     72 (transl. ibid., p. 84), etc.; on the topic of vision in the *Sūrsāgar* see (Hawley [1983] 1989, pp. 104–15).
[6]  Seeing the deity is a fundamental feature of rituals, as with the often-commented *darshan* of the deity. Still, cult-images
     kept in more or less accessible temples are far from being the only ones to be seen. The texts that tell Kṛṣṇa's life enhance
     the visual, even when the god is a flautist. As if they were painters or sculptors, poets reveled in giving details about the

related to the iconographical corpus seen in sculptures. The bulk of this corpus has been carefully collected and will be presented after an overview of the history of the motif in texts.[7]

The textual corpus that preceded the Bhakti poet-saints of North India who, from Mirabai (1498–1546) to Sūrdās, describe Kṛṣṇa as eating the butter he has stolen has long been identified. Canto 10.9 of the Bhāgavata-purāṇa, which focuses on the episode of the thief of butter, has been revealed as the first and foremost of their sources, if not the only one. As this Purāṇa has been an object of fierce debate regarding its date and place of composition, we refer the present reader to previous works for its South Indian origins. Its author(s) knew Tamil poems and were inspired by South Indian realities that were not accessible in northern India. Its composition in a deliberately archaic Sanskrit may have been started at the end of the ninth century and completed at the beginning of the 11th.[8]

These two limits are crucial to our understanding of the way the legend of Kṛṣṇa has been composed and diffused in the Indian peninsula, as well as in some parts of Southeast Asia. The Bhāgavata-purāṇa popularized some of Kṛṣṇa's pranks that were not part of the literature in Sanskrit or other North Indian languages predating this Purāṇa, but were alluded to in the Bālacarita, a play composed in South India around the seventh to ninth century—that is, at the beginning of the range of dates given for the composition of the Bhāgavata-purāṇa. The thief of butter appears in the Bālacarita, which revolves around the childhood of Kṛṣṇa, and it is vividly depicted in the Bhāgavata-purāṇa. In fact, all the earliest known texts evoking this mischief are from South India. How was such a connection established? One explanation concerns the fact that this trick falls within the childhood of the god or a part of his life that was not originally included in the earliest texts, and those sections were the most reworked and interpolated over the course of time in all parts of India.

The two fundamental texts in which the biography of Kṛṣṇa is delineated are the Mahābhārata and the Harivaṃśa, the latter being commonly defined along with the primary sources as a khila (appendix or "afterthought") of the Mahābhārata. The older Mahābhārata (fourth century BCE–fourth century CE) knows very little of the infancy of the god, while it is a main focus of the Harivaṃśa (second–third century CE). This distinction sanctions the recognition of a transformation from the Kṛṣṇa of the Mahābhārata into one of the many avatāras of the Hindu Viṣṇu—even if with a special status amongst the others—by the time of the Harivaṃśa. Kṛṣṇa is an adult figure in the epic, while the Harivaṃśa describes the birth, childhood, and adolescence of the god. In doing so, this text provides a biological model of the relation between the supreme deity of the Mahābhārata and the Kṛṣṇa of the Harivaṃśa, whose supplementary arms growing during his puberty provide a striking illustration of the process.[9]

Such a division of roles between the texts accounts for the absence of a Butter-thief in the Mahābhārata. Conversely, a child fond of butter seems ideal to be featured in the Harivaṃśa in which milk products are very present from the first mention of the cowherd's village where the child Kṛṣṇa is to be raised. In Chapter 49 of this text, 1498–1546, Nandagopa and Yaśodā, the foster parents of Kṛṣṇa, go back to their village with Kṛṣṇa as baby:

---

appearance of the god, while in festivals processions, plays, dance recitals, and the like, very rarely does the one who embodies Kṛṣṇa play the flute he holds. The flute is more a means of identification and a symbol of the relationship established with the devotee than an instrument that produces sound.

[7] (See Banerjee 1978, pp. 126, 131–32; Hawley [1983] 1989, pp. 52–95; Preciado-Solis 1984, pp. 53–56).

[8] The list of dates given by L. Rocher (1986, pp. 147–48) shows the difficulties of dating this text. The span of ninth–11th century given in this paper matches with that of most of the authors. The dating given in A. J. Gail (1969, p. 12) or P. Hacker (1959, pp. 121–28), according to whom the *Bhāgavata-purāṇa* would date to the eighth century, being considered obsolete by most scholars since the publication of F. Hardy (Hardy 1983). Hardy (Hardy 1983, pp. 485–88) recapitulated the issue before comparing the text with the Ālvārs' corpus to demonstrate their close link; he concluded that, most probably, the *Bhāgavata-purāṇa* dates to the end of the ninth or the beginning of the 10th century. Iconographical details suggest a date during the second half of the 10th century, at the earliest (see Schmid 2002, pp. 42–47; 2014, pp. 57–97).

[9] As a way to connect the two texts, there are many echoes with the *Mahābhārata* among the episodes of Kṛṣṇa's childhood in the *Harivaṃśa* in the form of visions of a future that was already known when this prequel was composed. (See Schmid 2010, pp. 59–76, 139–43, 397–408).

"There are many open spaces, happy and strong people are around; there are a lot of ropes in
the station and the sound of the churning is heard from every place.
Lots of buttermilk are around and the earth is wet with curd. Milkmaids create sound with
the noise coming from the churning sticks that they stir." (Harivaṃśa 49.24–25)[10]

In the following chapter (HV 50), Kṛṣṇa performed his first miraculous deed by upturning a cart.
Nandagopa sees the chariot lying upside-down together with many broken pots, presumably these
were vessels containing milk-products.[11] However, nowhere in the critical edition of the Harivaṃśa
is Kṛṣṇa's fondness for butter, or for yogurt, to be found—much less his theft of them. In chapter
51 that follows the two chapters in which milk-products are so present (49 and 50), even though the
infancy of Kṛṣṇa and his brother is vividly evoked[12]—the two boys play pranks on the cowherds in
their homes; they are covered with dust and other matter including cow dung, that will accompany
them till Sūrdās,[13]—but butter is not mentioned.

"Both had long arms that looked like snakes. They moved everywhere. With dust on their
bodies, they shone like baby elephants.
Sometimes their bodies were covered with ashes and sometimes with cow dung. They ran
around like the sons of fire god.
As they crawled on their knees in some places, they looked enchanting. They went to
cowsheds for playing and their bodies were covered with cow-dung.
Both were blessed with prosperity (śrī). They gave great pleasure to their parents. Here and
again, they would be mischievous with people and laugh.
Both roamed around the camp (vraja), doing various pranks. Nandagopa was unable to
control these two.
Once, Yaśodā became very angry with the lotus-eyed Kṛṣṇa. She brought him by the side of
a cart and she summoned him again and again." (Harivaṃśa 51.8–13)[14]

In this passage, the grammatical dual declension is extensively used for two brothers who are
defined as one body divided into two (HV 51.4). These two are so naughty that the foster parents
decide to get tough with them. But when Yaśodā does what she must, only Kṛṣṇa is mentioned. He is
tied to a mortar with which, eventually, he uproots a pair of arjuna trees. Here again one cannot find
any mention of butter in the critical edition of the Harivaṃśa. But that is not the case with the southern
versions of the text, in which, in between the stanzas 12 and 13 of this same chapter 51, the episode of
the theft of butter makes its appearance. The passage is developed in what became two appendices
in the critical edition (9 and 9A, 33 and 52 lines, respectively). Similarly, the southern versions of
the Mahābhārata itself mention the theft of butter by Kṛṣṇa (see Sabhaparvan, app. 1.21, l. 767–769),

---

10　*kṣamapracārabahulaṃ | hṛṣṭapuṣṭajanāyutam |*
　　*dāmanīprāyabahulaṃ | gargarodgāranisvanam ||49.24||*
　　*takranisrāvabahulaiḥ | dadhimaṇḍārdramṛttikam |*
　　X After 25ab, D6,T1.2.4,G1.2.4.5,M ins.: *navanīta+parikṣiptam | ājyagandhavibhūṣitam |49.25ab\*624||*
　　*manthānavalayodgārair | gopīnāṃ janitasvanam ||49.25||*
11　*sa dadarśa viparyastaṃ bhinnabhāṇḍaghaṭīghaṭam |*
　　*apāstadhūrvibhagnākṣaṃ śakaṭaṃ cakramāli vai ||50.13||*
　　"He saw the broken vessels and pots. The cart was lying upside-down, with the wheels up and its axle ruined."
12　*visarpantau tu sarvatra sarpabhogabhujāv ubhau |rejatuḥ pāṃsudigdhāṅgau dṛptau kalabhakāv iva ||51.7||*
13　As we can see in the poem cited at the beginning of this article, where Kṛṣṇa is marked with cow dung.
14　*kvacid bhasmapradigdhāṅgau karīṣaprokṣitau kvacit |*
　　*tau tatra paridhāvetāṃ kumārāv iva pāvakī ||51.8||*
　　*kvacij jānubhir uddhṛṣṭaiḥ sarpamāṇau virejatuḥ |krīḍantau vatsaśālāsu śakṛddigdhāṅgamūrdhajau ||51.9||*
　　*śuśubhāte śriyā juṣṭāv ānandajananau pituḥ |*
　　*janaṃ ca vipra kurvāṇau hasantau ca kvacit kvacit ||51.10|| ( … )*
　　*atiprasaktau tau dṛṣṭvā sarvavrajavicāriṇau |*
　　*nāśaknuvad vārayituṃ nandagopaḥ sudurmadau ||51.12||*
　　*tato yaśodā saṃkruddhā kṛṣṇaṃ kamalalocanam |*
　　*ānāyya śakaṭīmūlaṃ bhartsayantī punaḥ punaḥ ||51.13||*

making clear that the whole that comprises the Mahābhārata together with the Harivaṃśa has been modified in South India to include a series of mischievous acts in which butter plays a major role.

These southern versions of the Mahābhārata and Harivaṃśa are known through manuscripts written in Grantha or Telugu scripts and collected in the south of India. They are considered to be later than the critical edition for many convincing reasons. In the southern versions of an epic that consists of the Mahābhārata and its khila, Kṛṣṇa wanders from one house to the other, stealing milk, yogurt, and butter, breaking the pots where they are kept and sharing their contents with his friends. These narratives are very similar to the references to the butter-fondness of Kṛṣṇa at the end of Act 1 and the beginning of Act 3 of the Bālacarita[15] and to descriptions in chapters 10.8 and 9 of the Bhāgavata-purāṇa, in which the influence of the Divyaprabandham composed in Tamil is particularly conspicuous. In these southern accounts—whether they are composed in Tamil, Prākrit, or Sanskrit—Kṛṣṇa is tied to the mortar with which he uproots the arjuna trees, because he has stolen butter. The narrative thus presents a logic slightly different from that which is at work in the earliest northern versions of the Mahābhārata and Harivaṃśa. In those, Kṛṣṇa is punished because he was naughty; in the later, southern texts Kṛṣṇa is punished because of a specific misdeed, namely the theft of butter. The theft of butter is invoked in place of a series of pranks, in which it might have been considered as a grand finale. But the earliest Mahābhārata and Harivaṃśa are not the sole texts in which the butter-theft is missing. Furthermore, its absence in these earliest versions of the epic appears more problematic than it seems if we look towards the sculptural tradition. But first let us consider additional texts to see what additional clues they might provide.

Nothing is said about butter, butter-thievery, etc., in the Buddhist Ghata-jātaka (Jātakas no 454) telling the story of Kṛṣṇa as one of the previous lives of the Buddha in this collection of narratives belonging to the Pāli Tipitaka of the Theravādins. The full story is told in the commentary which was composed around the middle of the fifth century, by a Sinhalese monk.[16] Nor is anything about a butter lover mentioned in the Jain Antagaḍadasāo, the eight of the 12 Aṅgas of the Śvetāmbara canon of the composed in Prākrit (ardhamāgadhī) around the fifth century CE.[17] This text tells the story of Kṛṣṇa's brother Rāma, presented there as the eighth son of Devakī and a disciple of the Jain Tīrthaṅkara, Ariṣṭanemi. We also look in vain for the butter-theft in the Viṣṇu-purāṇa and the Brahma-purāṇa, which contain the earliest known Purāṇic version (the two texts are almost identical) of the childhood of Kṛṣṇa (fifth–sixth century). And, finally, nothing of that sort appears in the pierre de touche of the critical editions of Mahābhārata and Harivaṃśa, the Bhāratamañjarī, a versified summary of these two texts composed around the middle of the 11th century in Kāśmir.

According to this collection of texts, a fondness for butter was not a characteristic of Kṛṣṇa in the early period in northern India. The Sanskrit and Prākrit traditions from this area seem to ignore this episode until the spread of the Bhāgavata-purāṇa, after the 11th century. Still, there is in North India another type of evidence that may attest to the existence of a Butter-thief from at least the fifth century: Some of the first known representations of Kṛṣṇa as a child in sculptures.

## 3. First Sculptures: North India

The iconographical evolution of Kṛṣṇa parallels that of the texts. First, the Kṛṣṇa of the Mahābhārata is represented as a deity with four arms and his characteristic attributes of mace, disk, and conch, when depicted as the "ordinary form of the god" whom Arjuna longs to see after having been granted the

---

[15]　It has to be noted that after 13ab, the manuscripts from the south insert the following passage:
*uvāca śiśurūpeṇa carantaṃ jagataḥ prabhum |*
*ehi vatsa piba stanyaṃ durvoḍhuṃ mama saṃprati |*
*|etāvantam itaḥ kālaṃ kvā gato 'si gṛhād bahiḥ |*
*ity ādāya kare putraṃ gṛhān nirvāsya sā ruṣā |51.13ab\*642||*

[16]　(See Couture and Chojnacki 2014, pp. 129–47). This Jātaka certainly had a previous life of its own, from an oral version that I would presume to be located in the north of India, given the history of the narratives incorporated in the Jātaka corpus.

[17]　(Couture and Chojnacki 2014, pp. 165, 195).

vision of the cosmic deity in the Bhagavad-gītā.[18] During the second to fourth century CE, this iconography becomes identified as that of the Hindu Viṣṇu, a deity who is the source of all the others, including members of the Vṛṣṇi "family" of Kṛṣṇa (Schmid 1997, pp. 71–77; Couture and Schmid 2001). From around the Gupta period (fourth–sixth century), Kṛṣṇa is featured with the iconography corresponding to the story told in the Harivaṃśa as a child who fights demons with two bare hands and no weapons.[19] During the same time period, in regions where inscriptions dated in the Gupta era are found, the Hindu Viṣṇu emerges with some of his avatāras. Then, Kṛṣṇa takes center stage defeating the bird-ogress [Pūtanā], upturning a cart, fighting the nāga Kāliya, the ass demon Dhenuka,[20] the demonic Pralamba, the bull Ariṣṭa, the horse demon Keśin, lifting up Mount Govardhana—and, it can be considered, stealing butter.

Two stone reliefs depicting a child with one hand in a pot are known from two North Indian sites very distant from one another: Vārāṇasī (Uttar Pradesh, Figure 1) and Maṇḍor (Rajasthan, Figures 2–4).

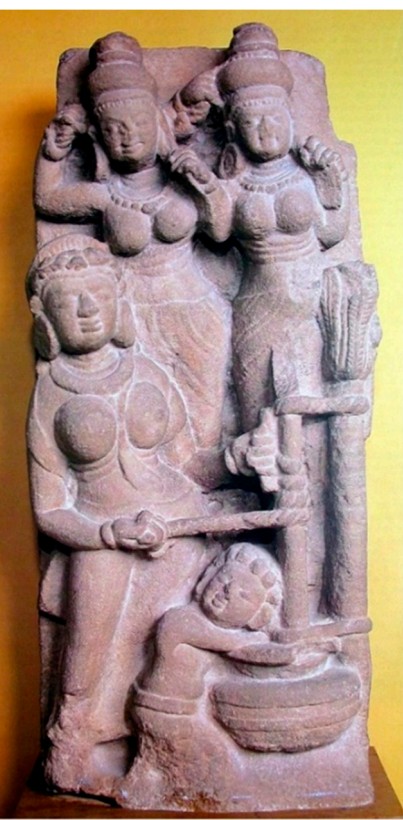

**Figure 1.** Churning of Butter, fourth–sixth century. Vārāṇasī, Uttar Pradesh. Sandstone. Bharat Kala Bhavan (BKB 180).

---

[18] *Bhagavadgītā* 11.45–46 (*Mahābhārata* 3.25–42).

[19] In very few cases the Kṛṣṇa of the *Mahābhārata*, the adult counterpart of the child Kṛṣṇa, is represented and then the iconography which is used is the same as the one of the Hindu Viṣṇu, the supreme, adult deity whose four arms and attributes are conspicuous characteristics (Schmid 1997, pp. 75–77).

[20] Balarāma is said to fight the ass demon Dhenuka and Pralamba, a demon of human appearance, in *Harivaṃśa* but it is not possible to distinguish between the exploits of Kṛṣṇa and the ones attributed in this text to his brother. Moreover, Balarāma's deeds, like the fight against the ass, are said to be Kṛṣṇa's works in some texts, including the *Mahābhārata* (Podzeit 1992; Schmid 2010, p. 290).

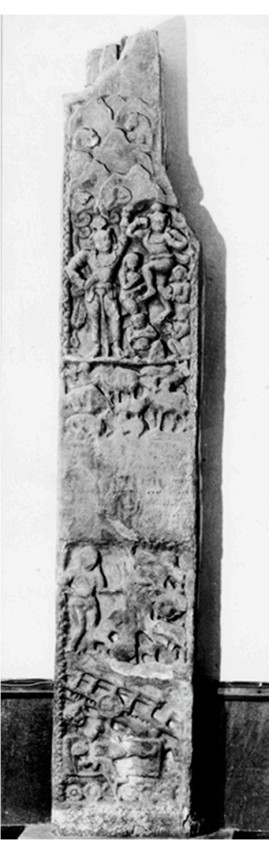

**Figure 2.** Doorjamb with Exploits of the Young Kṛṣṇa, fourth–sixth century. Maṇḍor, Rajasthan. Sandstone. Jodhpur Museum (photo by courtesy of the American Institute of Indian Studies).

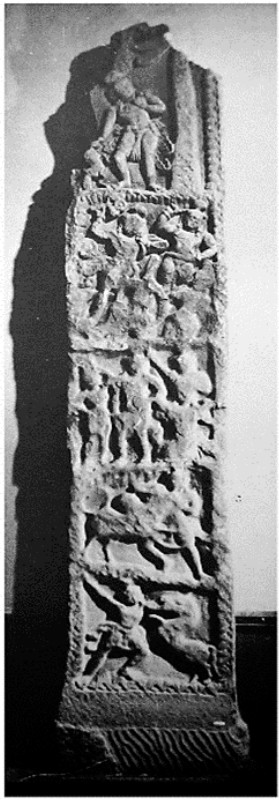

**Figure 3.** Doorjamb with Exploits of the Young Kṛṣṇa, fourth–sixth century. Maṇḍor, Rajasthan. Sandstone. Jodhpur Museum (photo by courtesy of the American Institute of Indian Studies).

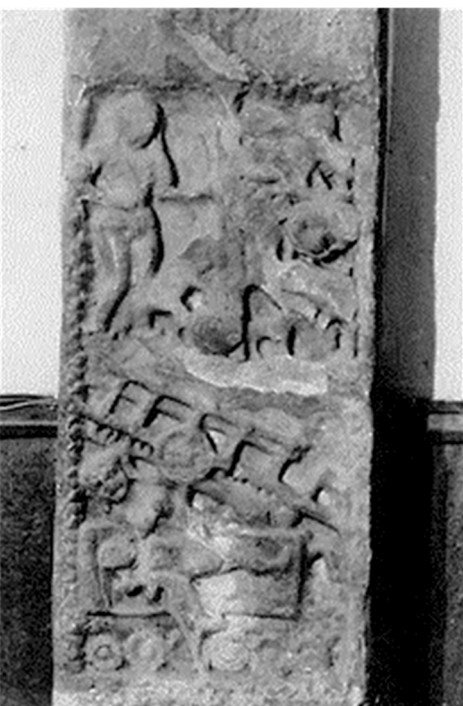

**Figure 4.** Detail of Figure 2: Churning of the Butter.

Identified as representations of the theft of butter, they present a strikingly similar iconography.[21] Dating between the end of the fourth and the sixth century, they appear to be older than the first known texts in which the motif emerges.

A woman stands on the left, facing the viewer. She is busy churning the contents of a big pot placed at her feet with a rope; below her legs, a kneeling child raising his face towards her and in the direction of the viewer, inserts a hand in the pot. On the Vārānasī fragment, two other women carry pots on their heads in the background; these women may also have been on the Maṇḍor relief (Figure 2), but the space above the churning scene is worn, and the carving is no longer discernible. Today, to the right of the churning scene, cows with their calves fill the space from the bottom to the top of the panel. On the Vārānasī piece, it is possible to see that the churning rope is attached to a tree, whose appearance is the same as the one given on contemporary or slightly later reliefs of the arjuna trees between which Kṛṣṇa crawls in the episode with the mortar.

The style of the two reliefs is not similar, nor are the techniques, as the Maṇḍor piece is carved in very low relief on a flat and tall stone, while the Vārānasī piece is much more deeply sculpted. The Maṇḍor scene is sculpted with two or three[22] other episodes of the Kṛṣṇa legend in panels on the same stone, and five other episodes featuring Kṛṣṇa as the hero appear on another stone with which it forms a pair (Figure 3). Viewed together, the whole ensemble of sculpted narrative panels confirms the identification of the butter-churning scene as related to the legend of Kṛṣṇa (Figure 4). These carvings present a coherent narrative cycle with Kṛṣṇa as a main protagonist; if the relief from Vārānasī had been part of such a cycle, the example illustrated in Figure 1 would be the sole scene to have survived.

The iconography of the two sculptures is so close it raises questions about the diffusion of the motif. Was it linked to lost texts? A woman churning while a young boy inserts his hand into the pot as others look on or milk cows constitutes the main element of the scene The child may be qualified as a kind of foreground, or as a detail of the main subject of the scene, which is the churning. The characters,

---

[21]   (Hawley [1983] 1989, p. 55).

[22]   The two episodes of the cart and of Pūtanā follow each other in the same chapter of the *Harivaṃśa*; they were gathered on the Maṇḍor relief, as well as in the contemporaneous cycle of carvings from Deogarh (Madhya Pradesh; Figure 5).

their attitude, etc., match the description of the cowherd village in the Harivaṃśa, where milkmaids churn away. However, there is no specific verse in the Harivaṃśa, the Viṣṇu-purāṇa/Brahma-purāṇa, nor in any text predating the Tamil sources that mentions a little boy. Since these two carvings are so similar, whatever their textual source of inspiration, be it oral or written, would have included a theft of butter from a pot placed on the ground, while a milkmaid is engaged in churning.

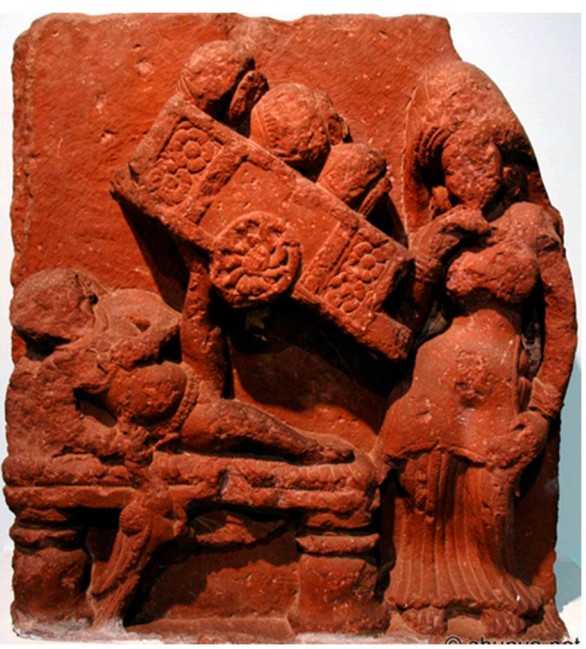

**Figure 5.** The Episode of the Cart and Pūtanā. Fifth–sixth century. Deogarh, Madhya Pradesh. National Museum, Delhi.

## 4. The Northern South: Karnataka

At the close of the period to which the North Indian reliefs are attributed, namely the end of the sixth century, the same iconographic formula started to be used in Karnataka, first in caves, then a little later at the built temples of Bādāmī and the nearby site of Paṭṭadakal. These two sites are situated in an area that represents a form of northern South, where North Indian culture, including Indo-Aryan texts, met a Dravidian realm. Bilingual inscriptions in Sanskrit and old Kannaḍa dated from around the late sixth or the early seventh century have been found in the same zone.[23] Many representations of Kṛṣṇa are found at these sites, which were under the control of the Cāḷukya dynasty at the time they were made. Exploits of a young Kṛṣṇa, one after the other, have been carved in narrative relief friezes on the lintels of Caves II and III of Bādāmī (Figure 6), on the base of a structural temple called the Upper Śivālaya of the same site (Figures 7 and 8), on pieces today reused in different parts of the same site (Figure 9), and on interior pillars of the Kāśiviśvanātha Temple at Paṭṭadakal (Figure 10).

In some of these friezes, a toddler is depicted inserting his hand in one pot churned by female figures, as part of a series of episodes that includes the death of Pūtanā, the overturning of the cart and the breaking of the trees (Figures 6, 7 and 10). In others, a simple churning, without the young boy, is figured (Figures 8 and 9). We will come back to this point. The chronology of the events followed in these series does not correspond exactly with the narrative in texts where the Butter-thief makes his appearance, whether they are composed in North Indian languages or in Tamil. Such a chronology is uncertain at Maṇḍor where the episode of the breaking of the trees is not represented (was it supposed

---

23　The Halmidi inscription is considered the earliest bilingual Sanskrit and Kannaḍa inscription (Mysore Archaeological Department 1938, pp. 72–74). It was found in the Hassan district in which are situated the sites of Bādāmī and Paṭṭadakal. One of the bilingual inscriptions was found in Bādāmī itself.

to be on the blank band of stone?), but in textual sources, Kṛṣṇa always (1) sucks the poisonous breast of Pūtanā, (2) steals the butter, before being (3) tied to the mortar with which (4) he uproots the arjuna trees. In the first carvings found in Karnataka, the episode implying butter can be depicted before the god sucks the poisonous breasts of Pūtanā.[24] Still, Pūtanā, the mortar and the uprooting are associated in the carved tradition as in the textual one. Such association requires further comment.

All the episodes from the childhood of Kṛṣṇa in the reliefs from Maṇḍor and the northern Gangetic plain of the fourth–sixth century and from sites in Karnataka of the late-sixth century correspond to one passage of the Harivaṃśa. For the period of time to which these carvings have been assigned, we have seen they are already rather numerous. In Karnataka, the Slaying of Pūtanā relates to different scenes than at Maṇḍor and Deogarh in Madhya Pradesh (Figure 5), where the Pūtanā episode is conflated with that of the cart. There, Kṛṣṇa is depicted lying on the cart when Pūtanā comes as a woman who becomes a bird that Kṛṣṇa strangles (Figure 5), or Pūtanā comes as a bird and becomes a woman whose breasts Kṛṣṇa sucks (Maṇḍor, Figure 4), as is told in the Harivaṃśa. At Paṭṭadakal, on the other hand, three discrete episodes are represented: The cart, a combat against a bird very different from the crow (śakunī) that is the form of Pūtanā in the Harivaṃśa and the earlier representations, and an ogress whose breasts Kṛṣṇa sucks. In fact, the episode of Pūtanā has been expanded there. Following the accounts composed in North Indian languages and the corresponding carvings, J. Hawley (Hawley 1987) has demonstrated how the story of the Harivaṃśa has been split into two in the Bhāgavata-purāṇa. In a previous publication, we have elaborated on the "shadow-motifs" that show how Tamil texts and stone representations of the far South are the missing link of the story (Schmid 2013).

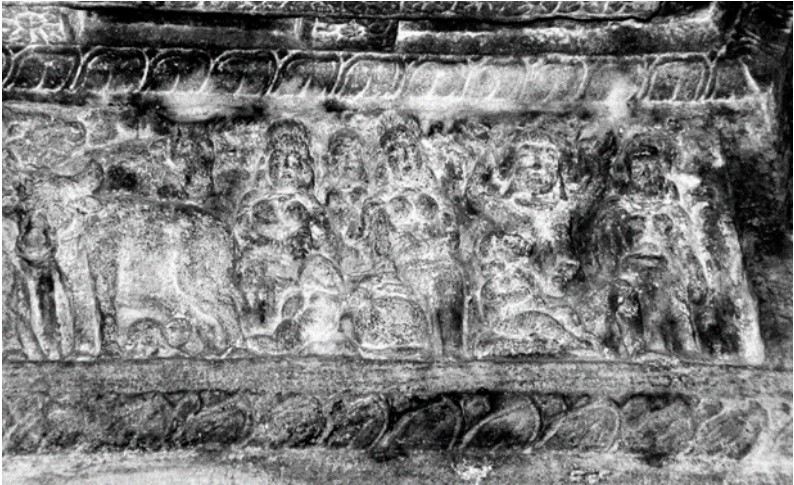

**Figure 6.** Exploits of the Young Kṛṣṇa (left to right): Churning with the "Butter-thief," Slaying of Pūtanā, end of the sixth century. Interior lintel, Cave II at Bādāmī, Karnataka.

The case of the Butter-thief is different from the Pūtanā episode, but it is analogous in terms of the presence of shadow-motifs, that result from transfers of an original story. The process of formation of the transfer-motif is this: One original motif is recorded in Sanskrit narratives; it is then split into two in Tamil texts. This split gives rise to a second motif distinct enough from its source to be considered a different one, even if, historically speaking, it is the "double" of one previously-born motif. The double-motifs of the legend of Kṛṣṇa were transmitted in Sanskrit with the Bhāgavata-purāṇa,

---

[24] The churning of butter is depicted before the Pūtanā episode according to the order of events that seems the more logical if we follow the texts. To the other side of the churning of butter is the exchange of babies or another scene alluding to the arrival of Kṛṣṇa to the cowherd village (Figures 6–9); in Figure 10 events are presented in an exploded perspective, but Pūtanā has been depicted just before or after the churning.

which drew inspiration from Tamil poetry. The heron "Baka" of the Bhāgavata-purāṇa outlined after the bird shape of the ogress Pūtanā in the early Harivaṃśa is one example of this process.

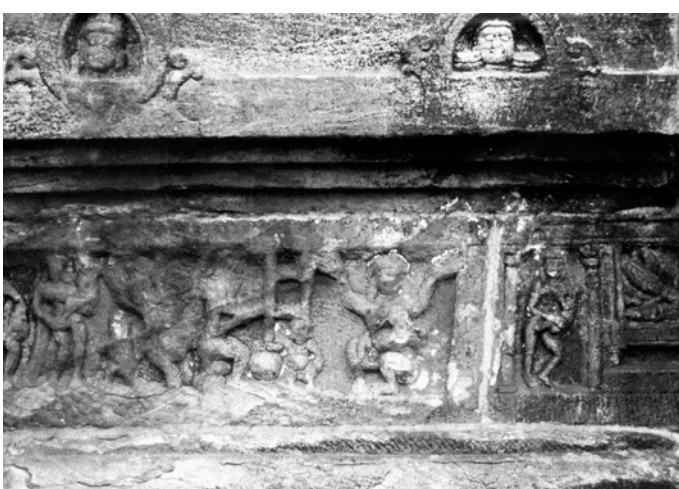

**Figure 7.** Exploits of the Young Kṛṣṇa (left to right): Arrival of Kṛṣṇa at the Cowherd Village, Churning with "the Butter-thief," Slaying of Pūtanā, sixth–seventh century. Base of the western side of the Upper Śivālaya, Bādāmī, Karnataka.

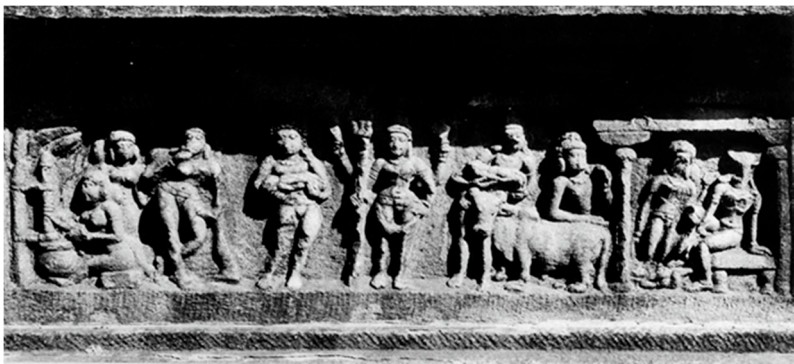

**Figure 8.** Exploits of the young Kṛṣṇa: Churning without the "Butter-thief," Arrival of Kṛṣṇa at the Cowherd Village, the Goddess, Transfer of the Baby to the Cowherd Village/Killing of Kṛṣṇa's Sister, sixth–seventh century. Base of the southern side of the Upper Śivālaya Temple, Bādāmī, Karnataka.

Iconographic formulas are key to such splits. On the one hand, from the Harivaṃśa to the Viṣṇu-purāṇa, the evolution of the narrative can be clearly followed with whole verses of the Harivaṃśa used in the Viṣṇu-purāṇa and with literary formulas common to these two texts. At the time that these texts were circulated, there were also iconographic formulas in circulation throughout a rather vast territory. Those visual patterns were used like textual stanzas: From one site to the other, the iconography exhibits variants, but in readily recognizable formulas. The churning of the butter as seen at Maṇḍor, Vārānasī, and northern Karnataka before the end of the sixth century is one such formula. With a child introducing his hand into a pot placed on the ground, this visual formula presents details that are not included in any earlier or contemporary known texts. These sculptures may express the creativity of a visual world that does not necessitate that we propose an underlying textual source. Before Tamil texts plausibly dated to the sixth or seventh century, no known text includes a description of Kṛṣṇa taking the content from a pot that was in the process of being churned. Still, those details of the images were part of the construction of the legend of Kṛṣṇa from Rajasthan in the north, to Vārānasī, eastwards (or the reverse) and north of Karnataka, to the south of the peninsula. At Bādāmī and Paṭṭadakal, the formula is included in iconographical series at sites where carvings may have been made in connection with South Indian texts or inspired by a tradition of carvings whose origin was situated in North India,

as indicated by the identification of the two early reliefs of Maṇḍor and Vārānasī with representations of a Butter-thief as early as the Gupta period.

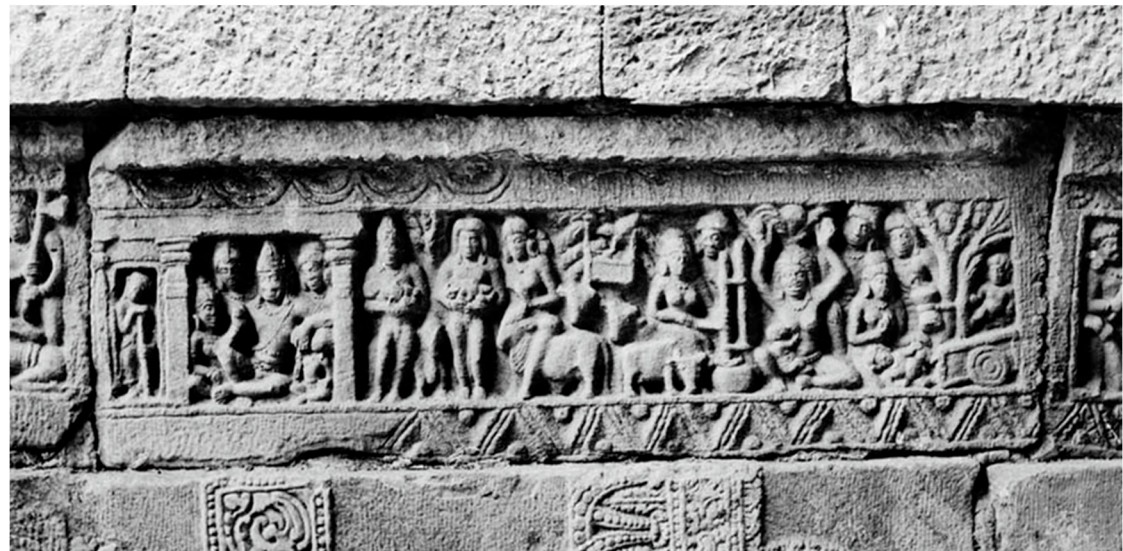

**Figure 9.** Exploits of the Young Kṛṣṇa (left to right): Arrival of Kṛṣṇa at the Cowherd Village, Churning without "Butter-thief," Slaying of Pūtanā, Arjuna Trees Episode, sixth–seventh century. Carving from Bādāmī Fort, possibly transferred from the Upper Sivalaya Temple, Bādāmī, Karnataka. Bādāmī Museum.

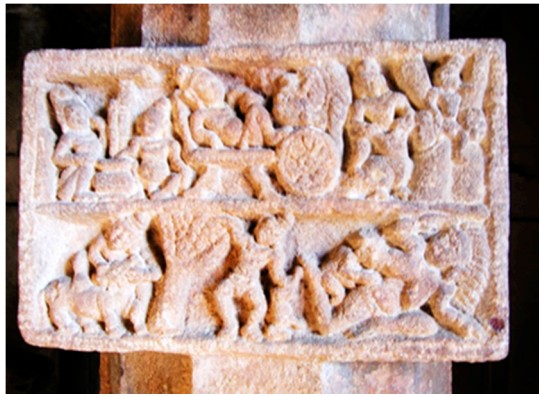

**Figure 10.** Exploits of the Young Kṛṣṇa (upper register, left to right): Churning with the "Butter-thief", Cart Episode, Fight against a Heron, Arjuna Trees Episode, eighth century. Kāśiviśvanātha Temple, Paṭṭadakal, Karnataka (photo by courtesy of Dominic Goodall, EFEO).

## 5. Carvings and Texts: Oral Transmission and Folk Culture

The issue of the difference between carvings and known texts might lead us to posit a theoretical "folk" culture with stories in North Indian languages that have not been preserved in any textual form. The stone carvings that have survived the ages would be considered evidence for their existence. Separate from the world of texts composed in Sanskrit (a language used by and for the elite), folk motifs in the northern and central areas of India would either have been introduced to South India where they influenced the development of the figure of Kṛṣṇa, or they would have been an ancient pan-Indian element. Associated with the world of herdsmen, the motif of a theft of butter would be but one of the themes of folk or "popular" (Hardy 1983) origin which found a way into texts of the elite first through Tamil compositions, more specifically Bhakti Tamil hymns. Composed in a vernacular and thus locally more accessible language, the Tamil poems would have been inspired by lost texts, of a similarly accessible, popular nature, whose existence may be attested by the stone carvings made from

the fourth century in the North. As most Tamil poems belong to a Bhakti corpus largely supposed to have been composed for a broad-based audience, and by authors said to cover a broad social gamut, they may well have been responsible for introducing folk themes into a religious corpus.

Even if some aspects of this hypothesis remain valid to our eyes, a critical examination of the first known stone representations raises several issues about their presumed textual underpinnings. Can stone carvings, usually patronized by rather well-off people, be considered clues to the existence of folk culture? Were the Bhakti Tamil hymns composed for a non-literate audience by poets following "popular" trends? Was the contrast between texts produced for and by an elite, be it Brahmanic, Buddhist, or Jain, and other texts associated with a folk world and supposed to have been transmitted orally so sharply delineated?

For our part, we are more inclined, with others, to think of the written form taken by texts in India as belonging to an environment where texts were recited, discussed, debated, and corrected as far as we know in the ancient world as they are still today. What is a "folk" world or a "popular" motif remains to be defined, but we doubt it is the source of the earliest evidence regarding the love of butter which underlies the Butter-thief figure. The first texts associating deities and the love for butter highlight links with literary and religious figures that are not considered to belong to a folk world, like the Vedic Agni, who under a child form devours the oblation of butter "with kisses on the ladle's mouth" in Ṛgveda 8.43,[25] and the Kṛṣṇa of the Bhagavad-gītā to whom Arjuna says: "With your firing mouths you lick all the worlds while devouring (them) . . . " (11.30ab).[26] There are links between the Vedic god of fire who devours the oblation and Kṛṣṇa in the Harivaṃśa; he is called Keśava, the "Hairy one," which is also one of Agni's names, compared to the son of the fire (HV 71.48–50), and qualified as god of the dark trail (kṛṣṇavartman) that is the fire itself, as has already been commented upon (Couture 1991, pp. 34–40). Such filiation reminds us that the concept of a divine child who eats butter and a god who devours everything, as manifest in the image of Kṛṣṇa stealing the butter, is deeply associated with a North-Indian cultural milieu. For this reason, it is unlikely that the figure of the Butter-thief in Tamil texts can be explained as having derived from a so-called folk background.

## 6. First Mentions of the Butter-Thief: Tamil Poems

The earliest known mention of a theft of butter by Kṛṣṇa is found in the Cilappatikāram (Cil.), usually called an "epic," and it presents striking parallels with a whole section of an amorous anthology of the Caṅkam corpus, the Kalittokai. These two Tamil poetical works are today considered to have been composed at a crucial time that saw the end of Caṅkam literature, which is the most ancient genre of Tamil literature, secular in nature—except for the Paripāṭal to which we will return later—and dating to the fourth century BCE to fifth century CE. This period also witnessed the emergence of the first Bhakti texts: The Śaiva Tēvāram and the Vaishnava Tiviyappirapantam, or Divyaprabandham, to use its more famed Sanskritized written title (TP; sixth–ninth century). Cilappatikāram is not considered part of the Caṅkam corpus, even if from a linguistic point of view it is closer to this corpus than to the Bhakti one. Its form, that of a long narrative poem drafted to demonstrate the power of karma in a Jain environment, reflects other concerns than the ones expressed in Caṅkam. Distinct from Caṅkam literature by their morphology, syntax, vocabulary, and meter, the Bhakti texts were composed in a very different spirit, as they are hymns in honor of deities who rarely appear in the Caṅkam corpus, other than the Paripāṭal. In the early Caṅkam anthologies, references to a Butter-thief motif are elusive and scarce; they become more particularized in the karma-inclined Cilappatikāram, where Kṛṣṇa is the one who steals the butter, whereas the later Divyaprabandham is laced with stanzas

---

[25]  *Ṛgveda* 8.43.10, transl. (Griffith 1973, p. 429).
[26]  *lelihyase grasamānaḥ samantāl*
  *lokān samagrān vadanair jvaladbhiḥ |*
  *tejobhir āpūrya jagat samagraṃ*
  *bhāsas tavogrāḥ pratapanti viṣṇo ||30||.*

in which a butter-loving Kṛṣṇa appears. Without presuming that Cilappatikāram is earlier than the Kalittokai, we first present its data as the motif is mentioned there in a way that helps to discern how it could be linked to the earlier Caṅkam anthologies. Dated to the seventh century at the latest, Canto 17 of Cilappatikāram tells of the child Kṛṣṇa, most plausibly for the first time in Tamil literature together with the Kalittokai, of which a presentation will follow.[27]

*6.1. Cilappatikāram*

The Cilappatikāram tells the story of Kaṇṇaki, the chaste wife who cursed the Pāṇḍya king for having had her husband Kovalaṉ killed, because he mistakenly held Kovalaṉ responsible for the disappearance of one anklet, a cilappu, after which the work got its title, The Story of the Anklet. The poem consists of three parts, each centered on one theme and one place, and thirty cantos, some of which focus on a deity, such as Indra, the goddess, Kṛṣṇa and Murukaṉ. The action of Canto 17 is staged in a cowherd village. Milkmaids, who have welcomed Kaṇṇaki on her way to see the Pāṇḍya king, sing in honor of Māyavaṉ or Māyōṉ, the Tamil name of Kṛṣṇa, one of the forms of Tirumāl, the Tamil Viṣṇu. In their songs they combine the churning of the ocean, the theft of butter and the three steps of Viṣṇu. The stanza where the god steals the butter from a pot occupies the central strophe of a cluster of three. The three of them are necessary to understand this crucial passage:

> Lord of the color of the sea! Once long ago you churned
> The belly of the sea with the northern mountain
> As a churning stick, and the serpent Vāsuki as a rope.
> Your hands that churned are the hands Yaśodā bound
> With her churning rope. Lord with a flowering lotus in your navel!
> Is this your māyā? Your ways are strange!

> He is the supreme being." Thus the immortals adored
> And praised you. You devoured the entire universe
> Though hunger does not trouble you. Your mouth
> That devoured is the mouth that licked the butter
> Stolen from the hanging pot. Lord with a wreath of rich basil!

> Is this your māyā? Your ways are strange![28]

> Tirumāl whom the host of gods adore
> And praise! You strode the three worlds
> With your two red-lotus feet to rid them of darkness!
> Your feet that strode are the feet that paced
> As the Pāṇḍavas's envoy. O Narasiṃha! Destroyer of foes!
> Is this your māyā! Your ways are strange![29]  (transl. [Parthasarathy [1993] 2004](), p. 177, emphasis added)

---

27  *Akanāṉuṟu* 59 features one Māl (Viṣṇu) playing on the banks of the Yamunā but this poem is a late addition to the anthology, of the same age than *Kalittokai* and *Paripāṭal* (Wilden 2018, pp. lviii–lix).

28  *vaṭavaraiyai mattu ākki vācukiyai nāṇ ākki*
    *kaṭal vaṇṇaṉ paṇṭu oru nāḷ kaṭal vayiṟu kalakkiṉaiyē*
    *kalakkiya kai acōtaiyār kaṭai kayiṟṟāl kaṭṭuṉ kai*
    *malark kamala untiyāy māyamō maruṭkaittē*
    *aṟu poruḷ ivaṉ eṉṟē amarar kaṇam toṟutu ētta*
    *uṟu paci oṉṟu iṉṟiyē ulaku aṭaiya uṇṭaṉaiyē*
    *uṇṭa vāy kaḷaviṉāṉ uri veṇṇey uṇṭavāy*
    *vaṉ tuḷāy mālaiyāy māyamō maruṭkaittē.*

29  *tiraṇṭu amarar toḻutu ēttum tirumāl niṉ ceṉ kamala*
    *iraṇṭu aṭiyāṉ muu ulakum iruḷ tīra naṭantaṉaiyē*
    *naṭanta aṭi pañcavarkkut tuutu āka naṭanta aṭi*
    *maṭaṅkalāy māṟu aṭṭāy māyamō maruṭkaittē Cil. 17.32–34.*

The first and the third stanzas allude to two cosmic deeds of the deity, the churning of the ocean and the Trivikrama avatāra of Viṣṇu. Between them, the stanza where the Butter-thief made his appearance alludes to a cosmic feast of the god who swallows the entire world. Each exploit finds an echo in the terrestrial realm. The churning of the cowherd echoes the one of the ocean, the butter eater replicates the vision of Arjuna in the Bhagavad-gītā, and the steps of Viṣṇu resonate with the paces of the adult Kṛṣṇa of the Mahābhārata. The levels of the cosmic and mundane deeds are intertwined in a process destined to become typical of the Tamil Bhakti corpus. Other correspondences are staged here. The adult form and the child form are paired in the first two stanzas, and the avatāra of Kṛṣṇa is the main topic of the three strophes. Also, the link established with Trivikrama as a dwarf who becomes a giant corresponds to an explanation of the passage of the child form to the adult one, of the terrestrial incarnated level to the cosmic one, that is explicitly drawn in the Harivaṃśa where the growth of the child Kṛṣṇa is paralleled with the divine growth of the dwarf form.[30] Belonging to these many levels, the butter churned in the cowherd village and eaten by Kṛṣṇa is also the amṛta churned from the ocean and devoured by the god.

These Cilappatikāram stanzas come at the end of the canto, but they were already foreshadowed at the beginning, when the cowherd's settlement is described in detail, with numerous allusions to curd, milk, butter, and churning. The cowherds have to supply butter to the Pāṇḍya king, but the churning turns to disaster:

> " . . . Today it is our turn (to give) butter" said Aiyai,
> having called her daughter, taking churning rope and churning-stick,
> There appears the old milkmaid.[31] ( . . . )
>
> In the pot the milk has not curdled,
> from the innocent eye of the globular humped bull, water comes, oozing,
> something terrible will come.
> The fragrant butter from the hanging pot (uṟi) has been eaten,
> it does not melt, it diminishes, ( . . . )
> In the hanging pot (uṟi), the butter does not melt![32]

The cowherds try to neutralize the bad omen of the milk that does not curdle, by calling the god who plays the flute with their songs and dances.

The churning functions as a hinge between the different levels of reality. It is associated first with the god summoned with a song who accompanies the dances, and then with "The king of kings, the Cēraṉ" said to have "churned the ocean" (Cil. 17.31.3–4), and it comes again in the group of three stanzas in which Kṛṣṇa eats the butter. It links the divine deeds ("Māyavaṉ has churned the ocean with the serpent-rope", Cil. 17.20.1), the model of the king (the Ceraṉ), and the everyday work of cowherds ("Your hands that churned are the hands Yaśodā bound/With her churning rope." Cil. 17.32.3–4). Two remarks here allow for following the evolution of the motif.

First, the localization of the episode in the South is clear thanks to the uṟi, this specific pot containing the butter that Kṛṣṇa licks. This type of vessel is typical of the Tamil South, and as such it is not represented in the earliest carvings from North India, nor in the early Karnataka ones. Second, the

---

30  Before Viṣṇu incarnates in the form of a baby (*śiśu*), Brahmā tells him: "You will have to grow like in days of yore you were able of a triple pace . . . " *HV* 45.39; the parallel is enhanced by the fact that Kāśyapa is the father of both the dwarf and Kṛṣṇa (under the form of Vāsudeva for the latter).

31  *neym muṟai namakku iṉṟu ām eṉṟu*
*aiyai taṉ makaḷaik kuuuy*
*kaṭai kayiṟum mattum koṇṭu*
*iṭai mutumakaḷ vantu tōṉṟum maṉ Cil.* 17.1.7–10.

32  *kuṭap pāl uṟaiyā kuvi imil ēṟṟiṉ* ( . . . )
*maṭak kaṇ nīr cōrum varuvatu oṉṟu uṇṭu* ( . . . ) *Cil.* 17.2.1–2.

stanzas express a relation between a Kṛṣṇa-child becoming an adult Viṣṇu that corresponds with a stage of evolution of Vaishnavism similar to what appears in the Harivaṃśa.

For the moment, we can conclude that the Tamil stanzas where the motif of the Butter-thief appears were carefully composed along preoccupations first found in the context of the Harivaṃśa, one text that emerges as a source of inspiration for this Cilappatikaram canto. But it is clear that many elements do not come from the Harivaṃśa, starting with a thief who steals what is inside an uṟi. Is it possible to find another inspiration for a motif that assembles butter and uṟi, thievery and a kind of hunger? Tamil texts contemporary to or pre-dating the composition of Cilappatikāram 17 may provide some explanation.[33]

### 6.2. Caṅkam Literature

During the period when the first Bhakti texts started to be composed, probably around the sixth to seventh century, the Caṅkam corpus began undergoing an important anthologization process, probably under the influence of an increasingly important written world. In the earliest stratum of the corpus available to us today, assigned to around the first to third century, butter was not specifically associated with Kṛṣṇa for one main reason: The rare allusions to Kṛṣṇa are cryptic and debated, while his brother Balarāma seems to have been popular, as is notable in the Paripāṭal, an anthology that belongs to the later stage of the Caṅkam and contains echoes of the Harivaṃśa.[34] Such scarcity of the elements associated with the legend of Kṛṣṇa in the early Caṅkam makes Canto 17 of Cilappatikāram, which is contemporary to or a little later than the late stage of the Caṅkam corpus (sixth century), the first known evidence of many of the "southern" motifs of this legend. These are episodes whose origin is situated in the south of India from which they spread, such as the stealing of the gopīs' clothes, or episodes that seem to exclusively belong to South India, such as the breaking of the kuruntu tree,[35] to take two instances that knew a very different destiny. These motifs are numerous enough in Cilappatikāram 17 to allow us to consider this text as one of the first examples of a narrative developed in a southern milieu devoted to Kṛṣṇa, distinct from the northern ones.[36] In such a configuration, the folk character of this milieu whose supposedly oral nature would allow for its texts to be constantly reworked and to possess an openness quality, distinct from the more secluded specificities of an elite literary corpus, is used to explain the almost complete absence of Kṛṣṇa in the Caṅkam corpus,[37] whose specificities would have prevented a folk Kṛṣṇa-bhakti movement from appearing on too literate a scene.[38] Following such reasoning, we find ourselves on the same trail as the one leading to folktales developed around the figure of Kṛṣṇa that inspired early carvings in northern India. The earliest known sculptures and the earliest known texts would be two different expressions of the same folk milieu.

By contrast, butter and churning do appear in the Caṅkam corpus. In the older romance anthologies and in the Puṟanāṉūṟu of the epic genre, of which some parts are earlier and some later, but overall quite safely attributable prior to the fifth century, butter connotes abundance and riches. It is a mark of prosperity, and as such it is desirable. The products of the churning of the ocean, among them Śrī herself, are related to a butter used to enhance beauty, to bathe with, to worship gods, and to

---

[33] From the fifth century onward, writing practices gradually became more common in a world where it was first essentially used for marking one's ownership or for commercial and accounting purposes, as can be deduced from the corpus of more than 800 Tamil-Brāhmī inscriptions on potsherds published between the end of the 19th century and today. See Mahadevan (2003) for a presentation of early Tamil epigraphy and the history of writing in this part of the Indian peninsula, to be complemented by Rajan (2015) for more recent discoveries and hypothesis.

[34] (Gros 1968, p. L; Hardy 1983, pp. 204–5; Schmid forthcoming).

[35] The *kuruntu* story may be alluded to in *Akanāṉūṟu* 59 according to Hardy (1983, pp. 193–96), but as a late addition to this anthology, to be connected to *Paripāṭal* (see *supra*, note 27). This poem seems to us being in between the northern *arjuna* tree motif and the *kuruntu* finally developed in the south. (Schmid 2014, pp. 48–53).

[36] (See Hardy 1983, p. 179; Hudson 2002, pp. 136–38).

[37] (See Hardy 1983, pp. 128, 172–83, 559, and appendix viii).

[38] According to H. J. H. Tieken (Tieken 2001, pp. 138–43; 2003), Caṅkam texts were invented to create a literary background for Bhakti poetry, at a time thus when these texts started to be composed, from the six–seventh century only.

celebrate.[39]  In Nāṟṟiṇai 40, a young mother takes a bath in ghee (butter);[40] the association of ghee/butter and the newborn is also found in Naṟriṇai 380 as one substance with which birth is celebrated.[41]  In the romances, butter appears in association with love, as could be expected. In Naṟriṇai 12, the churning is associated with the beginning of the day as in Cilappatikāram 17, to constitute a metaphor about disappearance and transformation given by love.[42]  This comes also in Nāṟṟiṇai 84 when the poet evokes the saltiness [that] … :

> … rises as within the heat of unfinished butter
> eaten by the churning stick in the earthen pot.[43]

In Kuṟuntokai 106, the heroine compares herself to fire into which ghee is poured in a metaphor of burning desire.[44]

Butter offers similar connotations in the war-themed Puṟanāṉūṟu, but there it has been adapted to the epic nature of the anthology. It is delicious and points toward riches and its brilliance. The hymnist of 384 praises the one who pours ghee "more freely than water"[45] while Puṟanāṉūṟu 65 (l. 2) tells the dire situation that accompanies the death of a ruler by lamenting over giant pots, lying on the ground, that "have forgotten how to hold the ghee."[46] There butter emerges as a substance first and foremost linked to celebrations of victorious kings "who pour more ghee than there is water, sacrifice more than there are numbers, [and] spread your fame on earth," and Brahmins who "under the glow of three fires offer ghee according to their secret rites."[47]

Thought to have been composed at the same period of time as the Kalittokai to which we will come, the Paripāṭal presents a different set of evidence, as this is not a romance or epic collection of poems but an anthology of hymns, some in honor of Tirumāl-Viṣṇu. One would expect a greater likelihood of encountering Kṛṣṇa. In fact, however, we meet him only once. With markedly Vedic references and the tightness of his association with Balarāma, the Vaishnava god of the Paripāṭal corresponds with the early Bhāgavata traces of North and Central India, for whom the adult god of the Mahābhārata

---

[39]  Similarly, in the Bhakti corpora butter is one substance of baths given to deities.  Inscriptions also kept echoing a practice which continues today to be a "real" one.

[40]  [ … ] when the weighty girl joined [her] moist lids,
With [her] soft body cooled with green oil, adorned
With moistness in an oil (ney, ghee) bath, adorned with white mustard,
While [her] son sleeps with [his] foster-mother, fragrant of new
Birth, on the bed with soft cloth spread out to have perfume (?),
—because one is born who bears the name of [his] elder.
*Naṟriṇai* 40. (Text and transl. Wilden 2008, pp. 138–39).

[41]  The same *ney* designates butter and ghee.
*Fumbling with ghee (ney) and incense,*
*[our] cloth is stained with dirt and [our] shoulders, embracing [our] son,*
*While sweet milk drizzles from [our] soft breasts with beauty spots*
*smell of the newly born.*
*Naṟriṇai* 380 (text and transl. Wilden 2008, pp. 818–19).  The same use of butter is found in the *Divyaprabandham* where cowherds and people from the mountains celebrate Kṛṣṇa's birth by covering themselves with butter and dancing, *infra*, note 71.

[42]  When the churning-stick's base thunders, in motion to clear the butter,
In the milk-mixture (?), [its] foot (?) diminished, eaten away but the cord,
In the pot filled to the brim, smelling of wood-apple,In the oncoming dawn that remains she would go,
[her] body disguised, on her leg the anklets unfastened (?) *Naṟriṇai* 12. (text and transl. Wilden 2008, pp. 82–83).

[43]  *cuṭumaṉ tacumpiṉ mattam tiṉra*
*piṟava veṇṇi urupp' iṭatt'aṉṉa*
*uvar eḻu kaḻari ōmaiyam kāṭṭu*
*Naṟriṇai* 84 (text and transl. Wilden 2008, pp. 226–27).

[44]  *Kuṟuntokai* 106.5–6. (Text and transl. Wilden 2010, pp. 290–91).

[45]  (Transl. Hart and Heifetz 2002, p. 225). See the butter that makes the spear to glitter in *Puṟanāṉūṟu* 95 or the one used to cook delicious dishes in *Puṟanāṉūṟu* 379.9, 596.

[46]  On the contrary the poet of 396 asks himself if he should sing of his fragrant rice with ghee poured upon it.

[47]  *Puṟanāṉūṟu* 166, 2 (Hart and Heifetz 2002, p. 108); see also *Puṟanāṉūṟu* 15 (Hart and Heifetz 2002, pp. 12–13), in which oblations rose rich in butter and other elements of the sacrifice.

was the foremost.[48] The childhood of Kṛṣṇa is not directly mentioned.[49] Its cowherd aspect may be discerned, however, with the word kōvalaṉ, "cowherd" in an enigmatic verse associating one of them with a pot and the one with a plow:

Right and left, pot and plow, cowherd and guardian. (Paripāṭal 3.83)

The butter is not mentioned, but the kōvalaṉ, the cowherd, is paired with the one of the plow, or Balarāma, here a guardian and the only deity with such tool/weapon, and is provided with a pot, as if it would be his characteristic attribute.[50] Is it a pot containing milk-products? We can only stress that a pot was considered characteristic enough to be the equivalent for Kṛṣṇa as is the plow for his brother Balarāma. As such, it could well be the uṟi of the Cilappatikāram. A pot, designated as kutam, is at the origin of the South Indian motif of the dance with pots that appears in the Cilappatikāram, is further developed in the Divyaprabandham, and often represented on Cōḻa-period temples from the end of the ninth century. No matter which evidence you privilege, it is clear that the pot had its importance in the southern legend of Kṛṣṇa. The one held by the cowherd, the kōvalaṉ of the Paripāṭal, may signify an early association with butter. It does not have a clear equivalent in the earliest, northern, versions of the legend of Kṛṣṇa. With "Fragment" I.64–71, the churning of the ocean is also part of the Paripāṭal. The deity takes the shape of the snake-rope used to churn the milk ocean and of the churner, or both the shape of Kṛṣṇa and his brother in the Sanskrit accounts of the myth.[51]

No clear profile emerges from this review. Butter is a mark of riches associated with rituals, one characteristic that cannot be considered a distinctively southern one, as clarified butter is one main element of sacrifices attached to Vedic traditions; butter is introduced in love poems with a sexual overtone, but echoes of the same kind are also reported from Vedic traditions.[52] It is not unmistakably linked with Kṛṣṇa even in the Paripāṭal. But by far the anthology of the Caṅkam corpus in which the butter is most present is the mullai section, the "jasmine" section, of the late love anthology, the Kalittokai. Cowherd settlements, milk-products, the uṟi, and Mayōṉ-Kṛṣṇa himself are set characteristics of the mullai tiṇai, the jasmine interior landscape of Tamil literature after which this section of the anthology in kali meters has been named.[53] The Kalittokai consists of a series of poems that provide a vivid backdrop to Cilappatikāram 17, and in the Kalittokai, butter plays many roles.

*6.3. Kalittokai*

For a summary of references to Kṛṣṇa in the Kalittokai, let us refer the reader to the unrivaled study of F. Hardy (Hardy 1983, pp. 183–93), who reports 16 instances. Most of them are concentrated in a group of poems that belongs to the mullai section of the anthology (Kalittokai 103–108). These are related to a bull-fight and a dance, similar to the one staged in Cilappatikāram 17, and on which we will now focus, as there is no mention of butter in the other references to the deity dispersed in other poems of the anthology.[54]

---

[48] (See Gros 1968, pp. L–LI; Schmid forthcoming).

[49] The only combat mentioned in the *Mahābhārata* is against Keśin, where, of course, it is not associated with childhood; it has a peculiar status and significance. (See Schmid 1999).

[50] For commentary on this verse (*iṭavala kuṭaval kōvala kaval*) (see Gros 1968, pp. L–LI; Hardy 1983, p. 205), for whom *kutam* (in *kuṭam aval*) may mean "west," but the rest of the line goes against this interpretation, which is proposed only in brackets by F. Hardy.

[51] (Text and translation, Gros 1968, pp. 144–49).

[52] For a general overview (Brereton and Jamison 2020, pp. 46–47); the authors note (p. 173) that it is surprising not to see the churning to appear in those texts where butter is such an important element of rituals.

[53] On the *tiṇai*, (See Ramanujan [1967] 1994; Takanobu 1995). On the relationship between *mullai tiṇai* and Kṛṣṇa (see Hardy 1983, pp. 157–67), though it is possible that the poems studied by F. Hardy as "*kuṟiñci tiṇai*" are actually oriented as *mullai*. We would venture to suggest that these poems belong to a stage when Kṛṣṇa as a youth became one of the incarnations of a Tamil deity of youth of whom, up until that time, Murukaṉ was the main embodiment.

[54] Convinced of the existence of a folk milieu where Kṛṣṇa's stories were elaborated, F. Hardy does not mention butter, much less note its significance.

In *Kalittokai* 106, built around a bull contest of the cowherds who dance kuravai and praise the Pāṇḍya king, butter is part of the cowherd equipment, and the uṟi is mentioned from the very beginning:

> The inhabitants of vast lands wet at the time the rainy season that made its appearance, the(se) cowherds of improper (vaḻūu) language, with their herds of cows, who make their sweet flute of lengthy koṉṟai to resonate, having balanced the earthen vessels of harmonious (icai) sound together with the hanging pots (uṟi)/tie (imiḻ) to fasten (icaittal)/the earthen vessels tied with ties ... (Kalittokai 106.1–5)

Later in the same poem churning is mentioned with the ornaments it produces on the shoulders of the heroine:

> "Hey my friend, the ornaments of my shoulders,
> the dots above, from the churning of yogurt
> in the milk,
> are destroyed through the embrace that shed blood
> of the one who takes the murderous bull ... ", (Kalittokai 106.37–39).[55]

In the poem 108, the heroine takes over the role of a milkmaid who goes selling milk-products used as metaphors for the amorous exchange with the hero calling her:

> "Passing by here, at the time of going home,
> having sold enough of your curdled milk,
> the weapon of your smile proclaiming your power, you struck my breath,
> hey you! hostile woman which fault did I commit towards you?" (Kalittokai 108.5–7).[56]

Similar lines come later in the poem to be more precise about the love situation, and the image of the thief is introduced in the stanza we cited at the beginning of this paper:

> "Having sold butter, at the time of the return–does one remember?—on that day,
> When the flowers of jasmine densely flourished, near the small river of one wood,
> With your eyes similar to tender young mangoes, my heart
> you took to bind (me) and rule on the battle-field: what if you are not but a thief of a unique kind?". (Kalittokai 108.26–29)[57]

The thief is the woman from a village where churning is going on in suitable time:

> The sound of the butter is heard close by, it is not a distant thing,
> if the village is near, if time is the middle of the day .... (Kalittokai 108.35–36)[58]

Here, butter is a marker of prosperity as in the other anthologies. But, above all, it is a marker of desire in a cowherd settlement described like the one of the Harivaṃśa. Kalittokai 110 is the culmination of the association of love with butter, when the heroine calls the hero:

---

55    *pal ūḻ tayir kaṭaiya tāaya puḷḷi mēl*
      *kol ēṟu koṇṭāṉ kuruti mayakkuṟa*
      *pullal em tōḻirku aṇi ō em kēḷ ē.*

56    *akal āṅkaṉ aḻai māṟi alamantu peyarum kāl*
      *nakai vallēṉ yāṉ eṉṟu eṉ uyirōṭu paṭai toṭṭa*
      *ikalāṭṭi niṉṉai evaṉ pilaittēṉ ellā yāṉ*

57    *aḻai māṟi peyar taruvāy aṟiti ō a ñāṉṟu*
      *taḷava malar tataintatu ōr kāṉa ciṟṟāṟṟu ayal*
      *iḷa mā kāy pōḻntaṉṉa kaṇṇiṉāl eṉ neñcam*
      *kaḷam ā koṇṭu āṇṭāy ōr kaḷviyai allai ō.*

58    Butter noise is heard, not as a distant thing,
      if the village is close by, when it is the middle of the day ...
      *veṇṇey teḷi kēṭkum aṉmaiyāl cēyttu aṉṟiaṉṉaṉittu ūr āyiṉ naṉpakal pōḻtu āyiṉ*

"—Hey you!
Were you looking for the pastoral beauties of the large and fenced village
as a medicine for the aching scorpion of your desire?
I came to you, sketched a smile, but you consider me as a woman easy to get!
Thinking I was as generous of my body as I was of white butter!". (Kalittokai 110.1–6)[59]

At this point to sell butter is interpreted by the hero as the promise of a gift of another kind, a consent to the desire the milkmaid has aroused. The metaphor is extended in the poem when the hero says of his heart:

"Like a rope attached to the churning pole of your beauty
my heart goes round and round . . . ".) (Kalittokai 110.10–11)[60]

and, eventually, considers his life as the sap exhausted by the churning that produced butter:

"Sufferings kept growing time after time,
Turned into the juice of churned butter,
my melted heart wants no other medicine
than the touch of (your) hand!". (Kalittokai 110.16–19)[61]

In these stanzas butter appears as an integral part of the cowherd setting and a result of a churning that expresses the torments of the lover's heart. These are a substance and an activity that link love—the main theme of early Caṅkam anthologies and of the Kalittokai—with the pastoral deity of the Harivaṃśa. The importance taken by both butter and churning in the Cilappatikāram canto dedicated to the god who presides over the mullai tiṇai is thus understandable. In Kalittokai, stealing butter was not recorded as a specific activity of the mullai tiṇai, nor of Kṛṣṇa. But there is definitely a role for butter as a symbol for what the hero is ready to take without permission, while in the Cilappatikāram butter is something that is stolen by Kṛṣṇa. But who is this thief when he is not Kṛṣṇa?

*6.4. Thief of the Heart*

In Cilappatikāram 17, a first larceny is alluded to with the mythical churning celebrated twice in the canto—and once, just before the theft of butter. When gods and demons churn the ocean of milk, indeed the amṛta that comes out is stolen by Garuḍa and then taken by Mohinī. When the child Kṛṣṇa steals the butter from Yaśodā's churning, he acts as these other forms of Viṣṇu. In the Divyaprabandham, the Āḻvārs make this theme explicit by equating the three steps taken by Viṣṇu in the Cilappatikāram with a theft of the earth after having mentioned the theft of butter:

---

[59] *kaṭi koḷ iruṅ kāppil pul iṉattu āyar*
*kuṭi toṟum nallārai vēṇṭuti ellā*
*iṭu tēḻ maruntu ō niṉ vēṭkai toṭutara*
*tuṉṉi tantāṅku ē nakai kuṟittu emmai*
*tiḻaittaṟku eḷiyam ā kaṇṭai aḷaikku eḷiyāḷ*
*veṇṇeykku um aṉṉaḷ eṉa koṇṭāy oṇṇutāl.*
A more literal translation could be as follows: "Hey you, having looked
for the nice ones of the shepherds of low tribe (living) in the fenced (village) that is large and densely populated,
—A medicine for the squeezing scorpion that your desire is? Having come to join (you), having given the smile I sketched, o you who having seen me of easy access for joining, alas you have taken (me) for a woman easy of access for butter, thinking she (behaves) similarly as (she did) for the white butter.

[60] *mattam piṇitta kayiṟu pōl niṉ nalam*
*cuṟṟi cuḻalum eṉ neñcu*, A more literal translation could be as follows: "Like a rope (*kayiṟu*) attached to a churning stick (*mattam*) (by) your beauty/Having turned around/moving here and there (*cuṟṟi*) my heart is tossed/whirls."

[61] *evvam mikutara em tiṟattu eññāṉṟum*
*ney kaṭai pāliṉ payaṉ yātum iṉṟu āki*
*kai tōyal māttirai allatu ceyti*
*aṟiyātu aḻittu eṉ uyir.* A more literal translation could be as follows: "Affliction (is) increasing on my side, all the time,/ Butter churning milk-like whatever juice it (my life) has then become,/ not knowing any other medicine but the curdling by your hand,/ my life is destroyed!".

> All the butter stored in pots he stole and ate,
> Like he stole the entire earth from Bali and ate (it) with his large belly,
> When as a tall dwarf he made a pact for three steps of land to stride … (Nammāḷvār,
> Tiruviruttam 91, [TP 2568])

Thievery appears thus as an essential activity of the god, and the theft of butter as one among other thefts. But the word kalavu used to style the theft of butter in Cilappatikāram is specific and enhances another aspect of the earthly raid that matches the background provided by Kalittokai. Love is in the air.

Kaḷavu is so specific a word it has given its title to one of the first known works on the akam (love) genre of Caṅkam literature, the Kaḷaviyal Akapporuḷ, "The story of stolen love," of Iraiyaṉār. This was composed, it seems, at the time the Bhakti genre was flourishing, between the sixth and the ninth centuries.[62] Variously translated as "secret love," "premarital love," "clandestine love," etc., when encountered in the akam genre, kaḷavu designates a theft of a specific kind, the one of a lover. Even if it can designate a "simple" theft, it remains a word imbued with the Caṅkam love background that is the literary horizon of Kalittokai and Cilappatikāram. It introduces a supplementary nuance in Kṛṣṇa's mischief and suggests the presence of the love hero of Caṅkam. In Cilappatikāram 17 this character is well present indeed, and the mention of a kaḷavu does not come as a surprise. Between the churning for the Pāṇḍya king and the songs to Kṛṣṇa, or at the heart of the canto, Kṛṣṇa, then anonymous as the Hero in the Caṅkam poems, steals the bangles, the clothes, and eventually the beauty (taiyal) of a no less anonymous Heroine. There the god acts like the heart thief of the Caṅkam. As the Heroine of Kuruntokai 25 puts it:

> "Only the thief was there, no one else.
> And if he should lie, what can I do?"
> Cilappatikāram 17.24 (2–3) relays the lament:
> How can we describe his presence who cheated
> The girl, who stole his heart, of her virtue and bangles? (Transl. Parthasarathy [1993] 2004,
> p. 174)

At the beginning of Kalittokai 114, we encounter the Hero pretending he is a child—while it is obvious he is not (ōri putalvaṉ aḷutaṉaṉ eṉpa ō, Kalittokai 114.2). Such behavior seems to foreshadow the Kṛṣṇa of the Divyaprabandham, another "fake" child who not only steals butter but also clothes, bangles, and more from the gopis in the frame of a theft of cowherdesses's clothes, which is another episode that is first found in the South Indian corpus. In between the love-hero of Kalittokai and Divyaprabandham stands the Butter-thief—and love-thief—of Cilappatikāram. To summarize, Cilappatikāram 17 makes use of a pattern that is also staged in the Caṅkam Kalittokai anthology, featuring Kṛṣṇa as the Hero who steals everything from the milkmaid, starting with her butter.[63] The stealing is inspired by an amorous love poetry already associated with the cowherd world in the Kalittokai where butter is a metaphor for love, produced by a churning of the Hero's heart, desirous of "something" owned by the milkmaid. The mullai section of the Kalittokai and Cilappatikāram 17 share many motifs,[64] and the amorous patterns of the Caṅkam corpus constitute a major source of inspiration for the Kṛṣṇa of the Cilappatikāram 17. Kalittokai reveals the role of Kṛṣṇa as butter-raider.

---

[62]  This word and its connotation are first known to us through the commentary of Nakkīraṉ eighth century. (See Buck and Paramasivam 1997).

[63]  See, for instance the following stanza by Tirumaṅkaiyāḷvār (5.2.3; TP 1360): "Having taken a child form, he ate curd and entered inside me, his slave, / (he who is) the unique one creeping like a heron thief who inside fields plunders *kayal* fishes—the one of Kūṭalūr!"

[64]  The bull fight to gain a young woman's hand, another motif said to be typical of the south of India, but also a shadow-motif inspired by the demonic bull Ariṣṭa, which in our opinion, is shared by the *mullai* section of *Kalittokai* and *Cilappatikāram* 17, for example (Edholm and Suneson 1972; Hardy 1983, appendix VIII and IX).

Consequently, the first textual appearances of the Butter-thief figure merge Kṛṣṇa as an avatāra of Viṣṇu, in line with the Sanskrit Harivaṃśa, with an avatar of the love hero of the Tamil Caṅkam. Butter thievery makes its appearance in Tamil texts as the outcome of a Kṛṣṇa-oriented "North Indian" (vaṭa) tradition[65] that already described in some detail a cow herding settlement, where the churning of butter was a main activity. It fits into the previous Sanskrit tradition in focusing on the churning that strengthens the ties between Kṛṣṇa and Visnu and, at the same time, between the Caṅkam love hero and Kṛṣṇa.

## 7. Jaina Versions of the Story

Several Jain texts composed between the end of the eighth and the 12th century mention Kṛṣṇa's penchant for butter when telling the story of the tīrthaṅkāra Ariṣṭanemi, who is the cousin of Kṛṣṇa and his brother Rāma in the Jain traditional accounts.

The earliest of these is the Harivaṃśapurāṇa of Jinasena, a digambara monk who composed his Sanskrit work around 783 in Gujarat.[66] Its chapter 35 evokes the childhood of Kṛṣṇa. In 35.42 comes the death of Kupūtanā, in whom we recognize the Pūtanā of the Harivaṃśa, killed by a child, who spends his time as follows:

> Jiṣṇu (Kṛṣṇa) spent his days and nights in sleeping, relaxing, crawling on
> his stomach, kicking, running continuously, babbling, and eating butter. (Harivaṃśapurāṇa 35.44)

Verse 44 tells how this child fights a chariot-demon, and, in 35.45, how he uproots twin trees where two demons have taken place. The events linked with Kṛṣṇa's love of butter are thus the same as in the southern versions of the Harivaṃśa, but there is no mention of a theft of the loved butter. Similarly, in the Riṭṭhaṇemicaru of Svayambhu, an apabhraṃśa text composed in the second half of the ninth century, Kṛṣṇa is said to eat butter when he defeats the demonic bird sent by Kaṃsa (5.6):

A little later in the courtyard of the camp, Hari was having in his hand freshly churned butter. At that moment, from the sky came another deity sent by Kaṃsa. She has taken the form of a crow and was croaking.

In the Harivaṃśa of Puṣpadanta, composed in apabhraṃśa between 959 and 965 at the court of the Rāṣṭrakūṭas, in Karnataka, the section 85.6–7 evokes a dust-covered Kṛṣṇa frolicking and playing with the churn, the milk-products and the butter, then:

Some other time, Kṛṣṇa saw one ball of butter and eats it as if it was the glory of Kaṃsa. This depiction takes place before the attack of demonic creatures, starting with Pūtanā, followed by a chariot and the twin arjuna trees. Finally, in the Triṣaṣṭiśalākāpuruṣacaritra (the story of 63 tīrthaṅkara) of Hemacandra, who belonged to the second half of the 12th century (1160–1172) and lived in Gujarat, the tale about the bull Ariṣṭanemi comprises the episode of the two arjuna trees (here, as one demon who tried to crush Kṛṣṇa). Kṛṣṇa went between the two trees with a mortar and uprooted them. At that moment he is said to be covered with dust (8.5, 137–141), as in the first part of Harivaṃśa 51 where the breaking of the arjuna trees is told; afterwards he is depicted as a naughty child who takes fresh butter when it is churned (8.5, 143).

It is problematic to recognize the Jain Kṛṣṇa as having inspired the deity of Cilappatikāram 17. First, the dating implies these Jain accounts from Karnataka and Gujarat would have been inspired by the Jain Tamil epic poem, instead of the other way around. Furthermore, given the brevity in the Cilappatikāram of the allusions to a legend of Kṛṣṇa, which is much more developed in these more northern-located Jain texts, the Cilappatikāram could not constitute the sole source for them. At the same time, all the numerous episodes of those Jain accounts have their equivalent in the Sanskrit

---

[65]   Here we use North India to refer to texts composed in Indo-Aryan languages as the Caṅkam and the Bhakti corpus do (*vata moḷi*, the northern language, the language from the North), but also, of course, to refer to the earliest known carvings.

[66]   For this section about the Jain texts (see Couture and Chojnacki 2014).

Harivaṃśa and/or the Viṣṇu-purāṇa/Brahma-purāṇa, with the conspicuous exception of the theft of butter. Conversely, several allusions in Cilappatikāram 17 to the legend of Kṛṣṇa cannot be understood without other South Indian texts—the Caṅkam love anthologies, among which Kalittokai is prominent, but also the Tamil Divyaprabandham and the Bālacarita, to which we will return in a moment. These elements, including a fight against several bulls (see supra, note 63), the pulling out of the kuruntu tree (citrus), and the figure of the flute player, do not have clear equivalents in the oldest Sanskrit texts devoted to Kṛṣṇa, Jain or otherwise. They are some of these double-motifs to which we have already referred, born in the south of India from the encounters between a northern Sanskrit and carved tradition, and a literary tradition of the South. Besides, from the eighth to the 12th century, the diffusion of Sanskrit and Prākrit texts in Gujarat and Karnataka is more probable than familiarity with the Cilappatikāram, whose impact in Gujarat seems to have been limited by the Tamil language used to compose the poem.

The sectarian aspect of the texts constitutes a final difficulty. These Prakrit and Sanskrit Jain texts are much more critical of Kṛṣṇa than the Cilappatikāram is. They help us to perceive in the Tamil poem an irony that adds one more level to its complexity. It can be considered that in Cilappatikāram 17 we start from cosmic deeds to land on futile everyday events, the cosmogonic churning of the ocean being reduced to a churning by an old lady in a cowherd village, for instance. But there is still a grandiose scope in the Tamil lines, while derision cannot be missed in the relevant passages of the other Jain works. In the Cilappatikāram the stupid Pāṇḍya king who dies is an incarnation of Kṛṣṇa, but the Cēra king is another figure of Kṛṣṇa, and one who churns the ocean. It would be difficult to consider such a statement as disparaging, since the Cēra king is said to be the Cilappatikāram's patron. Eventually in the Jain texts composed in North Indian languages, Kṛṣṇa does not steal the butter, even though, given the mocking context of the Jain accounts, it would have been tempting to present him as a robber. It thus seems to us likely that these medieval Jain texts of Gujarat and Karnataka were part of a tradition that originated mainly in a northern area of India.[67]

To summarize, from around the sixth century and probably a little later, a contrast emerges between texts from northern and southern India. In the South, Kṛṣṇa's love for butter is mentioned much earlier and more frequently than in the North. From the end of the eighth century, Sanskrit and Prākrit Jain texts composed in Gujarat and Karnataka also mention the young god's love for butter. They derived inspiration from northern versions, though it cannot be denied that, like the southern versions of the Harivaṃśa, they were also influenced by southern works, as exemplified by the Cilappatikāram. But what about sculptures of the episode of the theft of butter?

The Maṇḍor relief of a butter-lover made before the sixth century comes from Rajasthan, an area bordering Gujarat, while all the episodes evoked in the Jain texts are represented earlier in stone in an area where Indo-Aryan languages are prevalent. In the Dravidian part of India, it is only in the north of Karnataka that some of the first carved narrative cycles include a butter lover. But the representation of a butter theft can be questioned. Is it really a Butter-thief that is represented? Or "only" a butter-lover and a churning? The theme of a larceny is not that obvious in the said sculptures. The child does not hide himself to take the butter and the milkmaid does not prevent him from doing so, nor does she scold him. We have seen that the toddler lifts his head and turns it towards the churning woman, as if checking if she can see him. In secondary literature, before the influential book of J. S. Hawley (Hawley [1983] 1989), the early stone reliefs were regularly identified as "the churning."[68] The attitude of the young butter-lover can be thought to be provocative, but it does not suggest the secret implied

---

[67] (See Couture and Chojnacki 2014, pp. 165–92) who do not mention a Tamil tradition. Probably, it did not seem relevant to them in this case.

[68] See, for instance, the labels of Banerjee (1978) to the Figures 9, 11, 14 and 19 "*dadhi-manthana*"; still P. Banerjee uses the expression "*Krishna stealing butter*" about one of them, p. 138. In this work, Tamil Nadu is present only for the large relief of Mahābalipuram, on which the churning itself is not represented. On a relief dated to the 11th century from Bhubanesvar (Banerjee 1978, Figure 48) a milkmaid looks at a little boy tasting her butter with tenderness, while a male figure lovingly watches them. Since this piece has been carved after the theft of butter has appeared in texts, including the Sanskrit

in the kalavu of Cilappatikāram 17. In contrast, the earliest representations in which a theft is clearly depicted were not made until the ninth century, and they come first from the Tamil area.

## 8. South-Indian Texts and Carvings from Seventh to Ninth Century

Soon after the Cilappatikāram was composed, the legend of Kṛṣṇa appeared under a much more extended form in two works and languages. It emerged in Tamil in the four thousand verses of the Divyaprabandham of the Vaiṣṇava saint-poets (Āḻvārs) and in Sanskrit with the play called the Bālacarita (Story of the Child), two works dated between the seventh and ninth century. In these two, thievery of butter took another turn to be more closely associated with the legend of Kṛṣṇa. In the Divyaprabandham, the love for butter is clearly mentioned in more than 200 stanzas that are authored by eight out of the 12 poets, including three of the most prolific: Periyāḻvār, Nammāḻvār, and Tirumaṅkaiyāḻvār (whose work constitutes one third of the anthology), and it emerges as one of the most famous characteristics of the young Kṛṣṇa.[69] If each of the Āḻvārs entertains a specific relation with the Sanskrit corpus and the patterns elaborated in the Kalittokai and Cilappatikāram, and each of them performs with a voice of his own,[70] the theft of butter is common to most of them and is revealed as the emblematic prank of the god. The following stanza of the Tiruviruttam of Nammāḻvār gives an idea of the extent of this practice. The devotee asks the bees:

> Unite me
> With the flawless flower-like feet
> Of the one who stole butter, and was scolded
> For many such things
> The king of gods is my lord, o Bees. (Tiruviruttam 54 [TP 2531], transl. Venkatesan 2014, p. 52)

Most poems of the Diviyaprabandham are similarly structured in stanzas whose four lines lead to a tendency to present vignettes that can be enclosed in one strophe only. In the vignettes thus drawn, a narrative emerges that links a theft of butter to the rest of the legend of Kṛṣṇa. The voracious toddler eats the butter and steals it so often that Yaśodā ties him to the mortar he stands on to reach the hanging pot where the butter is kept; with this mortar the young deity uproots the marutam (the Tamil arjuna) trees between which he goes. Hence the butter-theft is linked with the powerful uprooter of the arjuna trees. As we have already seen, that narrative is inserted into the traditional account with verses added to the Mahābhārata and the Harivaṃśa in their southern versions. These additions are positioned just before the uprooting of the arjuna trees. The story is encountered twice in the Bhāgavata-purāṇa, in chapters 8 and 9 (BhP 10.8.30; 10.9.8), where it is the source for one major twin motif, in which Yaśodā is depicted as seeing the universe in Kṛṣṇa's mouth (see below):

For the [pots] hung out of reach he (Kṛṣṇa) devises a way by piling up things or turning over a mortar and then finds his way up to the content by making a hole in the pot. (10.8.30)

Standing on top of a mortar for spices, he (Kṛṣṇa) had turned over, looking around anxiously because of the stealing; from a hanging pot (he) to his pleasure handed a share of butter and other milk goodies out to a monkey; watching him from behind, she (Yaśodā) slowly approached her son. (10.9.8)

---

*Bhāgavata-purāṇa*, it shows that the tradition of depicting a churning without any implication of theft has continued for some time.

[69] Pēyāḻvār, Poikaiyāḻvār, Tiruppāṇāḻvār, Tirumaḻicaiyāḻvār, Periyāḻvār, Kulacēkara Āḻvār, Nammāḻvār, and Tirumaṅkaiyāḻvār feature the butter theft. More than 200 mentions are found in the *Divyaprabandham*.

[70] Āṇṭāḷ's work is an interesting case, as she is specifically associated with Periyāḻvār who plays on the motif so many times, while, if Āṇṭāḷ mentions the churning milkmaids and the poisonous milk of Pūtanā (*Tiruppāvai* 7, *TP* 480; *Nācciyār Tirumoḻi* 3.9, *TP* 532 . . . ), the butter-eater goes unnoticed in her works. In a sole stanza Āṇṭāḷ alludes to the tying to the mortar, but as an allusion to the uprooting of the trees without mentioning the butter, just like the *Harivaṃśa* (*Nācciyār Tirumoḻi* 12.8, *TP* 624). In Āṇṭāḷ's corpus butter is one element of rituals, to be offered to the god (see *Tiruppāvai* 27, *TP* 500 or *Nācciyār Tirumoḻi* 9.6, *TP* 592). *Nācciyār Tirumōḻi* 14.2 (*TP* 638) allows us to see how to go from one use of the butter to the other when, having been worshipped by cowherds celebrating the uplifting of the Govardhana, Kṛṣṇa is said to have a smelly odor of butter.

Then Yaśodā binds her son to the mortar, a part of the story to which Cilappatikāram 17 alludes in the stanza just before the one of the "Butter-thief":

> Your hands that churned are the hands Yaśodā bound
> With her churning rope . . .

Divyaprabandham proves more creative than the "epic" as, in the numerous stanzas of the Bhakti corpus, the binding and the eating of butter are detailed, various, put in correspondence one with the other, and give birth to other motifs as well. The stolen butter is commonly held in the uṟi, from which Kṛṣṇa licks butter in the Cilappatikāram. Since many strophes are built from formulas or include formulaic epithets, allusions to the liking of butter also include instances where an uṟi does not appear. Instead of the mortar on which Kṛṣṇa climbs, an upturned pot may appear.[71] Eventually, Kṛṣṇa is sometimes caught by Yaśodā, sometimes by a group of cowherdesses as in the Periya Tirumaṭal of Tirumaṅkaiyāḻvār (138–139; TP 2786), or an anonymous milkmaid as in the following stanza of Nammāḻvār:

> You who hold aloft disc and conch as weapons
> Stole butter then cried
> When the cowherd woman bound you with ropes,
> My lord what's left to say in my lament? (Tiruviruttam 86 (TP 2563))[72]

Cowherdesses in a group figure less often than a sole woman to catch the child, but their band is very present at the background, watching Kṛṣṇa with amazement, laughing, complaining, losing their temper, etc. The beating of Kṛṣṇa by a group of cowherdesses, an anonymous milkmaid, or his foster-mother appears often. It is not always mentioned, however; in any series, several of Kṛṣṇa's (or other forms of Tirumāl-Viṣṇu) deeds may be enumerated, some added, others expanded, etc.

The episode of the uprooting of the two trees is often mentioned together with the theft of butter, even in strophes where the mortar is not mentioned and, thus, appears tightly linked with the theft of butter.[73] The uprooting of trees also presents variants, some that are to be associated with this episode only (demons or not, type of trees . . . ), and others that are typical of South India, where the pulling up of the trees has inspired the "twin motif" of the uprooting of another tree, the kuruntu.[74] In addition to this, we notice that the order of the events is not as fixed in the Divyaprabandham as it is in the southern versions of the Harivaṃśa or in the Bhāgavata-purāṇa. Still, when the killing of Pūtanā and the uprooting of the trees are mentioned together with the theft of butter, they usually appear with Pūtanā in the first position, the theft of butter in the second, and the uprooting of the arjuna trees in the third, as they do in the southern versions of the Harivaṃśa.[75] This order is justified by the mortar used as a punishment for the theft, whereas it was used as a means for giving a halt to a generally mischievous behavior of the god in the earliest, northern, version of the Harivaṃśa. The link with the Pūtanā episode is commonly introduced by the act of eating (just as he has "eaten"

---

[71]    See Tirumaṅkaiyāḻvār, *Ciṟiyatirumaṭal* 33 (*TP* 2690) that reminds us of the importance of the association with a pot; in this regard see also Periyāḻvār 16 and 17 where the cowherds celebrate Kṛṣṇa's birth by emptying pots full of curd and butter, to dance on them, while the forest-dwellers dance having smeared themselves with butter.

[72]    See also Pēyāḻvār 91 (*TP* 2372); Kulacēkara Āḻvār 2.4 (*TP* 661) where the Lord is tied by an angry milkmaid because he has eaten curd, butter, and milk.

[73]    The same formulas appear in poems mentioning only the butter, or the ones mentioning the butter and the breaking of the trees (like *uṟi mēl vaitta veṇṇey viḻuṅki; veṇṇey viḻuṅka; veṇṇey kaḷavu; veṇṇey uṇṭu*).

[74]    In the Tamil texts, the uprooting of the trees gave birth to the uprooting of the citrus tree (*kuruntu*), already mentioned in the *Cilappatikāram* (17.21.1 *kollaiyañ cāraṟ kuruntocitta māyavaṉ*). In between the *Harivaṃśa*, in which the trees are kinds of godlings, and the *Bhāgavata-purāṇa*, in which they are demi-gods, the *arjuna* trees can be transformed into a sole, rather demonic, *kuruntu* (see Schmid 2013, p. 43); see also, *supra*, notes 26 and 34.

[75]    See, for instance, Pēyāḻvār 91 (TP 2372) where Kṛṣṇa devours the milk of the ogress, then the butter and, thus, Yaśodā ties him off.

Pūtanā, Kṛṣṇa eats the butter; Kṛṣṇa eats all kinds of milk-products, starting with Pūtanā's milk, etc.).[76] No narrative logic is provided in this case, and the association may come from the storyline of the Harivaṃśa in which the chapter devoted to Pūtanā (HV 50) is positioned just before the one of the arjuna trees (HV 51). The connection between the events is looser in this case, following the sequence of the Harivaṃśa narrative.

Thus, it is found that, with the Divyaprabandham, the episode of the Butter-thief becomes an integral part of the legend of Kṛṣṇa. There, it is placed at the beginning of the childhood of the god and associated with the uprooting of the marutam/arjuna trees. Kṛṣṇa climbs on the mortar to reach the uṟis. There is no mortar in the Cilappatikāram, in which the only mention of the twin trees episode comes in honor of the goddess of Canto 12 who is praised as another Kṛṣṇa who, incidentally, walked between the maruta trees. In the Divyaprabandham, the tying of Kṛṣṇa to the mortar allows for the episode to be linked to the narrative presented in the earlier Sanskrit corpus, as in the following stanza by Tirumaṅkai that ends a decade cadenced with the binding of Kṛṣṇa:

> That garland of Tamil songs (is) by Kaliyaṉ the king of Maṅkai of fertile fields,
> (destined) to the feet of the Lord who was bound to a mortar because one day he played with
> the white butter of the milkmaids,
> the one who filled his stomach with the butter, the curd and the milk of the hanging pot (uṟi)
> ... (Tirumaṅkaiyāḻvār 10.6.10; TP 1907)[77]

Tirumaṅkai is considered to be one of the latest authors of the corpus, but the same scheme is encountered, among others, in Pēyāḻvār (91; 2372), who is certainly one of the earliest. The variants sometimes appear inside compositions by the same Āḻvārs. They speak of an inventiveness at work that makes it plausible that the Divyaprabandham designed the narrative that was destined to become dominant in the Bhāgavata-purāṇa. Eventually, the story made famous through the Bhāgavata-purāṇa emerges with all its details. They include an important one that seems to go far beyond the Butter-thief, a world eaten, stolen, and spit out. The Kṛṣṇa of Cilappatikāram 17 is praised by immortals as devouring the universe:

> You devoured the entire universe
>
> Though hunger does not trouble you. Your mouth
> That devoured is the mouth that licked the butter
> Stolen from the hanging pot,
> has inspired a stanza like Tirumaṅkaiyāḻvār 10.7.3 (TP 1910):
> This mischievous small one, Ah girls!
> filled himself with the inside of the hanging pot,
> —white butter piled as a white mountain—,
> And feeling drowsy this thief finally fell into sleep!
> Look: his hands are all butter, his childish tummy as swollen as in days of yore with the
> seven worlds! poor me, what shall I do?

With such a stanza, it is clear that the gulping of butter of the Divyaprabandham echoes the devouring of the world envisioned in the Bhagavad-gītā. From the butter-licking of Cilappatikāram 17, comes a cosmic vision of the universe inside of the god. This is what Yaśodā sees at the end of chapter

---

[76] See Periyāḻvār 2.1 (*TP* 223), Tirumaḻicai Pirāṉ 37 (*TP* 788), Tirumaṅkaiyāḻvār *Periya Tirumōḻi* 3.10.9 (*TP* 1246), Poykaiyāḻvār 18 (*TP* 2099), etc.

[77] *niṉṟār mukappu-c ciṟitum niṉaiyāṉ vayiṟṟai niṟaippāṉ uṟi-ppāl tayir ney,aṉṟ(u) āycciyar veṇṇey viḷuṅki y-uralōṭ(u) āppuṇṭ(u) irunta perumāṉ aṭimēl,naṉṟ(u) āya tolcīr vayal maṅkaiyar kōṉ kaliyaṉ olicey-t tamiḻmālai vallār,eṉṟāṉum eytār iṭar iṉpam eyti imaiyōrkkum appāl cela-veytuvārē.* "Look at him, for having eaten the butter of the cowherdess, he has been fastened", (*aṉṟu, uṭaivaṉ kāṇmiṉ -iṉṟu aycciyarāl aḷai veṇṇey uṇṭu, āppuṇṭiruntavaṉē*, with a deleted *samdhi*) comes as a motto in the hymn that this stanza concludes ...

8 of Book 10 of the Bhāgavata-purāṇa 10 (10.8, 33–45). The episode of Yaśodā seeing the universe in Krsna's mouth is positioned between the two thefts of butter (10.8, 29–32 and 10.9, 1–10). Having stolen butter, Kṛṣṇa ate dirt, revealed as this earth in which he rolled himself in a tradition we know from the early Harivaṃśa.[78] Desiring to see what he put in his mouth, Yaśodā sees:

> . . . the entire world, together with the moving and stable entities, outer space, and the sky, the directions along with mountains, islands, oceans, the surface of the earth, together with the wind, fire, moon and stars. She saw the circle of the planets, water, brightness (tejas), outer space, and also sky [ . . . ]

The theft of the butter introduces into Kṛṣṇa's childhood the mytheme of the cosmic child who contains the universe.[79] It has a Vedic background, and the Divyaprabandham, drawing an equivalence between butter and earth, makes use of a shared emphasis on the divine appetite. Just as he steals butter, the god steals the earth. Nammāḻvār prays to this thief eating the butter he has stolen, as, when previously, he took the form of a dwarf, then of a giant to steal the earth from Bali and eat its whole with his large belly (Nammāḻvār Tiruviruttam 91, supra, p. 22). In the Divyaprabandham the thievery of butter adds one more level to the parallel of the magic growth of the Vedic avatāra of Viṣṇu and Kṛṣṇa that was already stated in the Harivaṃśa. The motif is the object of further elaboration in the Divyaprabandham, in which, finally, the god spits out the world he has swallowed.[80] Such elaboration proves the genuineness of the theme in a Tamil environment—but cannot but be alluded to, in passing, as the study of this last variant is beyond the scope of this paper.

The number of stanzas where the Butter-thief appears in the Divyaprabandham[81] and the echoes provided in this corpus to famous mythological deeds of which new variants are given show the importance of the Divyaprabandham in the amplification of the Butter-thief motif. The position given to this episode in the Tamil Bhakti anthology speaks in favor of its South Indian origin. Similarly, the Sanskrit and Prākrit Bālacarita seems to present the successive additions that have been made to the "Story of the Child," since the time of the constantly reworked Harivaṃśa. The Bālacarita offers one of the variants encountered in the Divyaprabandham and not the version given in the Bhāgavata-purāṇa. The third act starts with a prelude where an old cowherd, speaking in Prākrit, depicts the miraculous events witnessed in a cowherd village where "no sooner milk was given by the cows than there was butter . . . "[82] The baby has been attacked by a demoness called Pūtanā who tried to murder him with her poisonous breast, and Kṛṣṇa killed her. Then, when he was only one month old, angry milkmaids came to tell Nandagopa's wife that her son was going from house to house to drink milk, entering one

---

[78]　The eating of the dirt seems to us as having been inspired by the dust, namely, the substance of earth, that covers Kṛṣṇa while the young deity plays when being scolded and bound to a mortar in *Harivaṃśa* 51. 7–9. In the *Divyaprabandham*, mud and butter are two associated substances that cover the body of the god and that he eats.

[79]　Such passages, of both the *Divyaprabandham* and the *Bhāgavata-purāṇa*, made the devouring Kṛṣṇa of the BhG echo the Epic and Puranic episode of Mārkaṇḍeya, who explores the world inside the body of Nārāyaṇa after having entered the mouth of a sleeping child lying on a banyan leaf who is none other than Kṛṣṇa; on this episode. (See Brinkhaus 2000; Couture 2007, and Schmid forthcoming).

[80]　See, for instance, the following stanza of Poykaiyāḻvār:
"Tirumāl,
Lord who became sky and fire,
The restless sea and the wind,
Lord sweet as honey and milk,
How could you hope to fill your belly
With the cowherd-woman's butter,
When you swallowed the whole earth
And spewed it up long ago?" (Poykaiyāḻvār *92, TP 2173*; transl. Cutler 1987, p. 126).

[81]　The approximate number of stanzas mentioning the theme given previously (200, *supra*, note 68) is an approximate statistical count based on the use of formulas; when one begins studying the motif, it appears that many more stanzas are linked with it.

[82]　A. Couture (Couture 1992, p. 129) gives the parallel passage in the southern versions of the *Harivaṃśa* where "since Kṛṣṇa is born, there is as much ghee as milk."

to devour the yogurt, gulping fresh butter in another, eating milk rice in a third one, and in yet another one was waiting to steal the pot of butter-milk.

So it was told by the angry cowherdesses to Nanda's wife, and angry, Nanda's wife took a rope and tied one end around his middle and the other to a mortar; seeing that the mortar moved, he threw it on a pair of demons called "Yamalārjuna."

The larceny is a logical conclusion to the previous behavior of the god who loves milk products and, most of all, butter. The mortar punishes gluttony and thefts, as in the Harivaṃśa, but now it is used to punish Kṛṣṇa for pranks that include the stealing of butter. Yet, as in some of the stanzas of the Divyaprabandham, Kṛṣṇa does not climb on the mortar, he has to deal with cowherdesses in a group as well as with Yaśodā, and fights against trees whose demonic nature is reminiscent of the evil kuruntu of the Tamil poems (and of some malevolent goddesses encountered in the Jain accounts). The account used in the Bālacarita presents common points with both the early Sanskrit tradition and the one developed in the Tamil texts.

But, if the Tamil corpus played so important a role, up to the composition of influential texts composed in Indo-Aryan languages such as the Bālacarita, and later the Bhāgavata-purāṇa, one would expect South Indian carvings to have depicted a Butter-thief—the more so, since it was so iconic in North India. However, carvings representing a Butter-thief in the Tamil country do not appear before the ninth century. They became more common from the 10th century. From the second part of the 10th and 11th century they are more numerous, but then they feature details that refer to the Bhāgavata-purāṇa, like the monkeys with whom Kṛṣṇa shares the butter he has stolen.[83]

How to interpret this gap between texts and sculptures? While the beginnings of the motif are marked by a gap of time between early carvings and texts in the north of India, an inverted gap marks its development in the South where texts come first and carvings follow with a substantial delay. We are left with the conundrum of North Indian sculpted representations corresponding with a Tamil textual tradition, while southern carvings correspond with Sanskrit texts. Will an alternative way of seeing the constitution of the legend of Kṛṣṇa in Tamil texts be possible by considering the visual tradition as one media for transmission?

## 9. The Churning

Actually, we look in vain for textual correlations to the North Indian and Deccani images among Sanskrit and Prakrit texts, and, furthermore, they do not match either the Cilappatikāram or the Divyaprabandham stanzas. There is no hanging pot in the earliest carvings; Kṛṣṇa does not lick the butter running down from an uṟi in either Maṇḍor or Vārānasī, and he does not behave this way in the first Karnataka carvings. The uṟi from which Kṛṣṇa steals butter is a product of a phenomenon of localization of the motif that has not been represented in the sculptural tradition until the second part of the ninth century. The first representations are found in the Tamil country and were produced during a period of time that logically associates them with the Divyaprabandham and not with the Cilappatikāram. Their devotional context, on Śaiva and Vaiṣṇava Bhakti temples, also favors the Divyaprabandham as a source of inspiration, rather than the Jain Cilappatikāram. Moreover, in the Śaiva temples, Kṛṣṇa is shown perched on the mortar that is not mentioned in the Cilappatikāram, but is important in the dominant variant of the story found in the Divyaprabandham (infra, p. 34, Figures 11 and 12).

---

83  (See Schmid 2002, pp. 45–46; 2014, pp. 60–61).

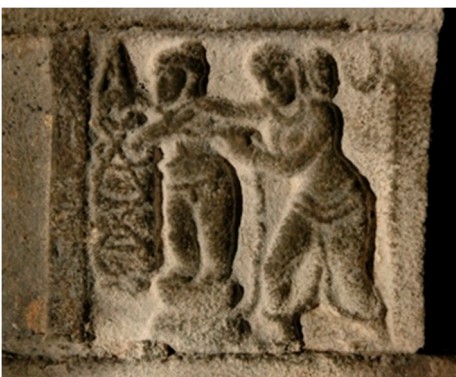

**Figure 11.** Kṛṣṇa climbed upon the mortar, Yaśodā comes to punish him, end of the ninth–beginning of the 10th century. Granite. Tiruvaṭutuṟai, Tamil Nadu.

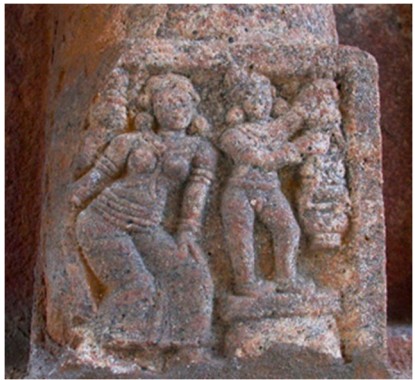

**Figure 12.** Kṛṣṇa climbed upon the mortar, Yaśodā comes to punish him, 10th century. Granite. Tirumaṅkalam, Tamil Nadu.

To sum up, during the first stage of the development of the legend of Kṛṣṇa, during the first millennium and whether we are in the North or the South of the Indian peninsula, there is no clear correspondence between carvings and texts produced at the same period and in the same area. Did those two domains develop autonomous traditions with but few contacts between them? If that is so, how and when did the contacts intensify? Did the intensification happen only with the emergence of the specific purāṇa that is the Bhāgavata-purāṇa? But before concluding with simply invoking the traditional hiatus between textual and visual data, let us reexamine a passage in the Harivaṃśa. When we reach the cowherd's settlement, this text invites us to pursue another trail:

> There are many open spaces, happy and strong people are around.
> There are a lot of ropes in the station and the sound of the churning is heard from every place.
> Lots of buttermilk are around and the earth is wet with curd.
> Milkmaids create sound with the noise coming from the churning sticks that they stir.
> (Harivaṃśa 49.24–25)

Here, the churning is used to depict the atmosphere of the cowherd village. To give a picture of such a camp in stone, a sculptor could similarly pick up this activity, portraying milkmaids busy with churning. To signify we are in this specific location where Kṛṣṇa is welcome, the presence of a little boy can be seen as rather natural. The depiction of an infant sets the scene of a cowherd village where a child is raised, and this may provide an intermediary between the tale of the texts and the depiction of the sculptures.

It is our contention that the early images were designed to portray the cowherd village described in the Harivaṃśa, even if no precise detail allows for a sure identification of the little boy of the carvings. The setting of the Maṇḍor stele supports the hypothesis. The scene where a child is depicted with

his hand in the churning pot occupies the center of the stele. Above this scene, a band has been left blank, unfinished, as though the sculptor was wondering what to depict at this place. On the top of it, the Govardhana episode is represented, and below the scene of the churning we see the chariot on which rests Pūtanā (under two shapes, woman and bird) whose life is sucked out by a baby Kṛṣṇa. There is no clear separation between the churning and this cart-cum-Pūtanā episode. On the paired stele, a thin band of stone demarcates each of the five different episodes, presented in succession from the top.[84] On the stele of the churning, the large band of stone left blank between the Govardhana episode and the churning interrupts the narrative. What was planned to be represented there? Or written? Whatever the answer, this band highlights the continuum depicted between the churning and the cart episode that are separated from the other scenes. It seems these two were designed to be part of the same sight, the churning being the background of the chariot-cum-Pūtanā episode, as it is in the Harivaṃśa and the Viṣṇu-purāṇa. It may have also been conceived as a background to the Govardhana episode, as in the representations in Karnataka and, later, in Tamil Nadu, but there was a difficulty in linking all the episodes on the stele.[85] The lack of models may have played a role in this difficulty, as we know of no other linked narratives of the Kṛṣṇa story in stone of that age. In any case, the activity of churning allows us to link all the scenes depicted on the Maṇḍor doorjamb as the devotional landscape of the first part of Kṛṣṇa's childhood. The Govardhana episode always had a specific status in this legend,[86] and the blank band of stone may also indicate its importance by allowing it to be highlighted on the whole of the stele and, in any case, presenting it as separate from the other depictions of the story of Kṛṣṇa.

To our eyes, the churning of the central part of the doorjamb was carved to represent the location of the cowherd village, in a visual formula that happens to connect various episodes to each other. Such a formula, created in North India, set the scene, marked the place where Kṛṣṇa performed his miraculous deeds as a child. When the legend made its way south, this visual formula remained the fixed background of several episodes. It was sometimes, in a conspicuous instance, included in the often represented Govardhana one. The figure of Balarāma, who is represented on Kṛṣṇa's right side in the carvings from Maṇḍor (Figure 3), Bādāmī (Figure 13), and farther south, later, at the Vaikuṇṭha Perumāḷ in Kāñcīpuram from the beginning of the eighth century (Figure 14) allows us to be sure that the Govardhanadhara background was conceived after the establishment of the iconographical formula in northern regions.[87] The churning milkmaids carved on the left of the Govardhanadhara at Bādāmī on the south wall of the upper Śivālaya temple are similar to the ones found in Vārāṇasī and Maṇḍor (Figure 13).

To the left of the Govardhanadhara at Bādāmī, the churning is represented in too similar a way as it is at Maṇḍor, more than one thousand kilometers to the north, not to have been sculpted from the same formula. Further south and a little later,[88] milkmaids likewise populate the background of the Govardhana relief at Mahābalipuram where a woman carrying the famous uṟi is depicted together with many other inhabitants and accoutrements of a typical cowherd settlement (Figure 15). As it has been demonstrated, the relief at Mahābalipuram constitutes a synthesis between the lifting of the

---

84 From the top of this doorjamb, we see the Dhenuka episode, the Kāliya, the Pralamba, the Ariṣṭa, and the Keśin ones, that is in the order given in the *Harivaṃśa* but for Dhenuka and Kāliya whose positions were exchanged. The murder of the horse Keśin is the last of the exploits of Kṛṣṇa before he goes to the city of Mathurā, and it marks the end of the deeds of the god as a child.

85 It is possible that the fragmentary carving of Vārāṇasī was also associated with a Govardhanadhara, as the only other Kṛṣṇa-devoted piece found in Vārāṇasī for this period of time is once again a Govardhanadhara (for an ill. Banerjee 1978, Figure 48). It is tempting to associate the two in the light of the other representation of the Govardhana episode.

86 The Govardhana story is told in three chapters of the *Harivaṃśa*, while other exploits are told in only one chapter or half a chapter, but for the Kāliya episode, which is told in two chapters. These two deeds have special significance in Kṛṣṇa's legend and are also, by far, the most often represented in carvings.

87 From one site to another from the sixth to eighth century, Balarāma wears the same headdress and costume; he similarly lifts his proper right hand as if willing to help his brother support the mountain or paying him an admiring tribute.

88 The dates of the rock-cut remains at Mahābalipuram are still debated; in agreement with most authors we think the Govardhanadhara relief was done at the end of the end of the sixth or at the beginning of the seventh century.

mountain from the early North Indian texts and aspects derived from Caṅkam, as seen in the flautist marking the center of the scene.[89] The renewed pattern of the churning perfectly fits such a synthesis.

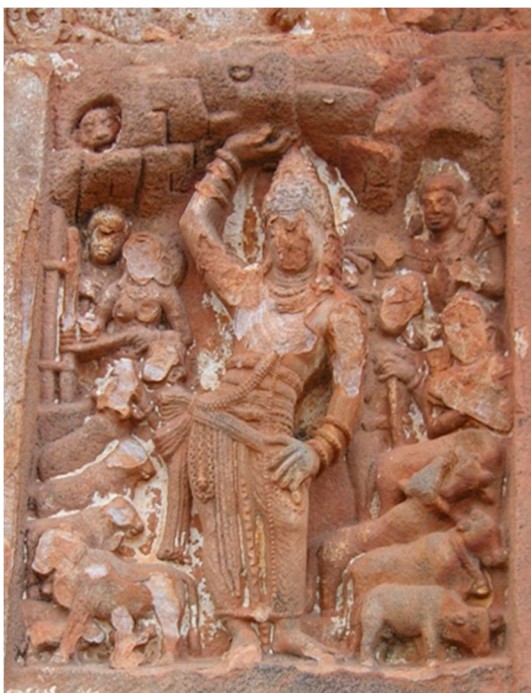

**Figure 13.** Govardhanadhara, sixth–seventh century. Southern wall, Upper Śivālaya, Bādāmī, Karnataka (photo by courtesy of Dominic Goodall, EFEO).

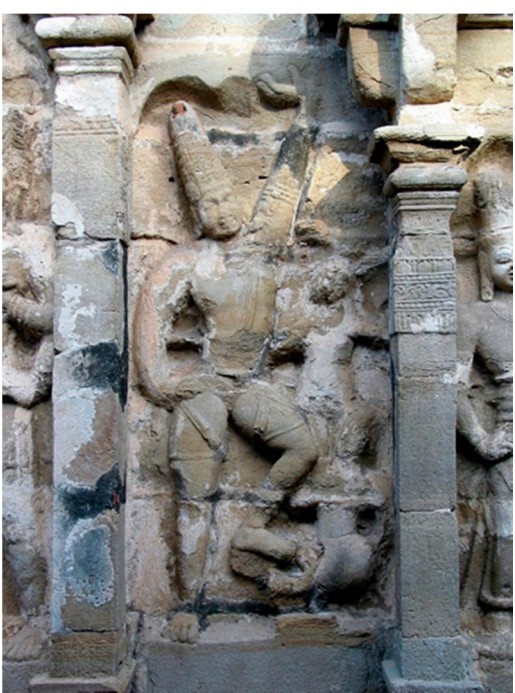

**Figure 14.** Govardhanadhara, beginning of the eighth century. Eastern wall, Vaikuṇṭha Perumāḷ, Kāñcīpuram, Tamil Nadu.

---

[89] (See Schmid 2014, pp. 43–56).

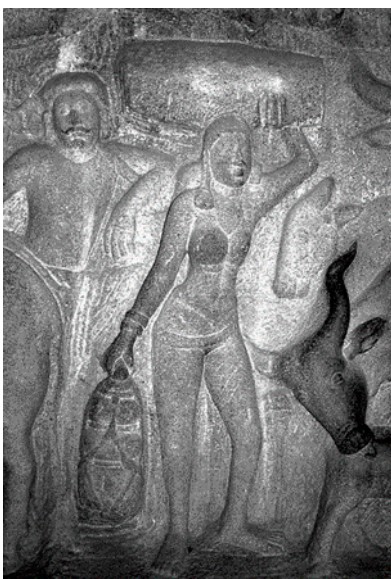

**Figure 15.** Detail of the Govardhanadhara Relief: Woman carrying uṛis, end of the sixth century. Rock-cut cliff. Kṛṣṇa Maṇḍapa, Mahābalipuram, Tamil Nadu.

We assume that in travelling to the south of India, the iconographic formula of the cowherd village underwent a specific transformation. The representation of the churning including a little boy whose presence indicates we are in Kṛṣṇa's village starts to be cleaved into two different scenes in the Karnataka representations. Churning is then considered as the background for two distinct scenes: The beginning of the childhood of Kṛṣṇa when the boy plays in the village and the Govardhana episode that marks the beginning of the divine growth or the end of the infancy-childhood cycle. Two series of carvings of Kṛṣṇa's childhood still visible at Bādāmī support this hypothesis. The churning is represented on them, before the Pūtanā episode, as usual, but without the little boy (Figures 8 and 9). These reliefs show that the toddler can be separated from a churning in which no theft of butter is implied. Here the churning serves as the indicator of the setting that was the backdrop of Kṛṣṇa's youth, as in the Govardhanadhara of the nearby temple of the Upper Śivālaya. In other friezes presenting Kṛṣṇa's life at Bādāmī, in cave II and on the base of the Upper Śivālaya itself, the churning is represented with the little boy when in proximity to the episode of Pūtanā, as it was in Maṇḍor (Figures 6 and 7).

The set of representations from Karnataka attest that visual formulas were accompanying the spread of texts. In the process of travel of the constantly reworked legend, phenomena occur that philologists know well: Elements of representations were pieced together while others were expanded or dislocated. The churning/cowherd village that constitutes the background of several episodes of the legend unites different scenes on the stele of Maṇḍor, but it is itself later on and in other areas split into several pieces. The fate of this pattern indicates that the visual formulas through which the legend of Kṛṣṇa was spread were composed of elements that could be combined in different ways. There were other visual elements to make clear the story takes place in a cowherd's station, like the cows visible on the left of Figures 6 and 8. Groups of cows are often depicted on the right of the representation of the exchange of the babies—that is the part of the legend when Kṛṣṇa is introduced in the cowherd's environment.

Moreover, from Karnataka and southward the transmission of texts passed through linguistically distinct realms in which the visible, concrete images traveling through their own specific channels, might have assumed a different importance. It is not by chance that in Karnataka the visual formula of the churning as developed in the North reappears under the same form in several series illustrating the childhood of Kṛṣṇa, but starts to be used in a different way with the Govardhana episode and other series. There, the boy disappears from the churning scene. Once one of the original components of the visual churning formula in North India, the toddler experienced another destiny in the Dravidian

world. He was there elevated to the status of the hero of an episode, the theft of the butter, that is told in Tamil texts.

We posit that the iconographic formula of the cowherds' village gave birth to the motif of an infant Kṛṣṇa stealing butter from a pot in the written texts. This development appears to have resulted from the isolation of one significant element of the early images in the process of building a legend that drew upon both visual and textual sources. At the time when the Kṛṣṇa legend was being translated into Dravidian languages, it acquired new patterns as a part of the process of translation-adaptation. That process and the necessity of being adapted to a very different literary environment can even involve some simple misunderstandings. The texts underwent a process that was accompanied and nurtured by the visual tradition, which was also another means by which the story was transmitted, travelling with the artisans themselves and the material they use as bases for their work as well as under the forms of portable images.

The boy represented in the frame of the churning became a Butter-thief elaborated on the model of the thief of the heart of the Caṅkam tradition. After the Pallava sites of Mahābalipuram and Kāñcīpuram, comes the site of Tiruveḷḷaṟai, farther south near Trichy. The base of the temple of this site can be dated to the eighth century, with one churning milkmaid on a panel, who looks towards a paired panel, in which Kṛṣṇa dances (Figure 16). There is no theft represented, only the joy of a little boy, the same who is later represented in numerous works under this exact posture but with a ball of butter in hand. We would venture that this type of representation corresponds to the stage of elaboration of a Butter-thief in texts, where a jubilant toddler first associated with the churning in carvings assumes more and more importance.

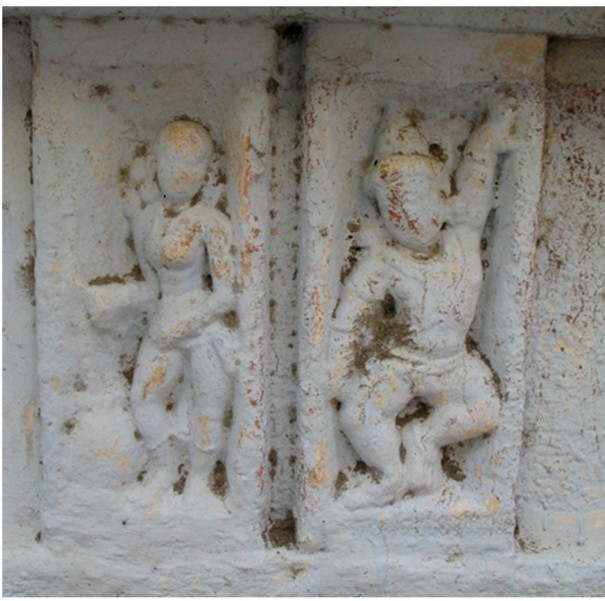

**Figure 16.** Kṛṣṇa dancing under the eye of one churning milkmaid.

From the end of the ninth century onward, Kṛṣṇa is represented stealing the butter in a suspended pot that he reaches by climbing up on a mortar in a series of reliefs carved on the bases of Cōḷa-period temples, as at Tiruvaṭutuṟai (Figure 11), Tirumaṅkalam (Figure 12), Tirukkoṭikāval, and later on as at Tirupuvaṉai (Schmid 2002, Figure 22), for instance. All the details told in the Divyaprabandham are featured. One angry woman, probably Yaśodā, comes to tie the young god to the mortar on which he has climbed (Figures 11 and 12). The Butter-thief starts his career as an anonymous child who forms part of the setting for Kṛṣṇa's exploits in a bucolic cowherd village with milkmaids churning butter. From the visual world in the North, the imagery and the stories migrated to Karnataka. There, the two schemes are sometimes combined with the boy inserting his hand in a pot placed on the ground, like on the earlier "northerner" representations, while another one climbs to reach a hanging pot, like the

later "Tamilian" figures as, for instance, at Somnathpur, after the middle of the 13th century (Figure 17). Eventually, the thief reached the whole of India, most probably together with the Bhāgavata-purāṇa.

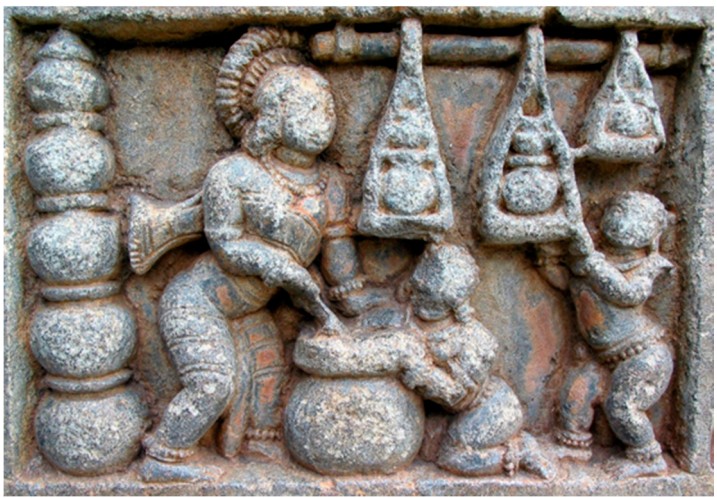

**Figure 17.** Two Boys Taking Butter (Kṛṣṇa and Balarāma?), one in a pot on the ground, one in an uṟi, 13th century. Keśava Temple, Somnathpur, Karnataka.

## 10. Conclusions, a Media as Important as Texts

At the end of this survey, we are led to think that, in the southern part of the peninsula, patterns according to which stone carvings were done have been a source of inspiration for the composition of texts in Tamil. The author(s) of the Cilappatikāram knew the texts in Sanskrit, Prākrit, Pāli, and maybe other texts as well, but we hope to have given reasons to suppose they were also aware of the traditional ways of representing a child Kṛṣṇa in the visual world. With the stone tradition we know from the site of Bādāmī, with the early Sanskrit texts that are the Harivaṃśa and the Viṣṇu-purāṇa, with the Caṅkam corpus, the poets started to feature a character they may not have been conscious of creating as it was already present in the visual tradition. This character was suggested by the very setting for the Kṛṣṇa story, the cowherd settlement, a landscape so ubiquitous that it gave birth to new episodes of his legend.

We would like to highlight the many ways for one legend to travel, including northern texts, southern texts, and visual imagery. The stories of Kṛṣṇa went to the south of the peninsula where they were localized with the help of a strong literary tradition having its own codes about a Butter-thief. There emerged a child who made his appearance as an antecedent of the adult god of the Mahābhārata and of the adult heart thief of the Caṅkam corpus. Comparisons among the representations of the Govardhana episode sculpted in Rajasthan, Madhya Pradesh, Karnataka, and Tamil Nadu show that the patterns of the reliefs produced in the north of India journeyed as formulas to the Tamil country. Just as texts, the visual world produced codes of representation and formulas that were subjected to the phenomena of combination, "conflation," evolution, and isolation of motifs. Texts and images were engaged in a constant dialogue which we only access through fragments. We should not miss these faint traces however, if only because they warrant another constant dialogue, the one of North and South in the Indian peninsula.

It is not the least bit paradoxical that the Butter-thief theme has become known in North Indian corpora like the Sūrsāgar that were used to enhance the literary value of the Hindi language. The makhan cor was born of an endless circulation inside the whole of a subcontinent, and this

heart-thief, neñcu kalvaṉ, was indeed worth being promoted to one valuable representative of a kind of "Indianity."[90]

**Funding:** This research received no external funding.

**Conflicts of Interest:** The author declares no conflict of interest.

## Primary References

Akanāṉuṟu

Wilden, Eva. 2018. *A Critical Edition and an Annotated Translation of the Akanāṉuṟu, Part. I—Introduction, Invocation—50*. Pondicherry: École française d'Extrême-Orient/Institut Français de Pondichéry.

Cilappatikāram

*Cilappatikāram*. [1892] 2001. *Cilappatikāram Iḷaṅkōvaṭikaḷaruḷiceyta cilappatikāramūlamum arumpatavuraiyum aṭiyārkku-nallāruraiyum*. U. Vē. Cāmiṉātaiyar Ceṉṉai: U. Vē. Cāmiṉātaiyar nūl nilaiyam. First published 1982.

Divyaprabandham

*Divyaprabandham*. [1993] 2002. *Nālāyirativviyappirapantam, Four Thousand Hymns of Twelve Alwars and Commentary by Dr. S. Jagathratchagan. English rendered from the Sacred book by Sri. Rama Bharati*. Chennai: Āḻvārkaḷāyvumaiyam. First published 1993.

Harivaṃśa

*Harivaṃśa*. 1969–71. *Harivaṃśa*. Edited by Vishnu S. Sukthankar, S. K. Belvakar and Parashuram Lakshman Vaidya. Poona: Bhandarkar Oriental Institute, 1969–71.

Kalittokai

*Kalittokai*. [1943] 1999. *Kalittokai Maturaiyāciriyar Pārattuvāsi Nac¬ciṉārkkiṉiyar Uraiyuṭaṉ*. Chennai: South Indian Saiva Siddhanta Works Publishing Society. First published 1943.

*Kalittokai*. 2015. *Kalittokai, Mulamum Naccinarkkiniyar uraiyum, Cempatippu, Tokuti 1 and 2. Rajeswari, T. (Critical Texts of Cankam Literature 3.1 et 3.2)*. 2 vols. Pondichéry: EFEO/Tamilman Patippakam.

Kuṟuntokai

Wilden, Eva. 2010. *Kuṟuntokai, Critical Edition and Annotated Translation of the Kuṟuntokai + Glossary and Statistics*. Critical Texts of Caṅkam Literature 2.1–2.3. 3 vols. Chennai: EFEO/Tamilmann Patippakam.

Mahābhārata

*Mahābhārata*. 1933–63. Mahābhārata, critical edition by Vishnu Sitaram Sukthankar, Poona: Bhandarkar Oriental Institute.

Naṟṟiṇai

Wilden, Eva. 2008. *Critical Edition and Annotated Translation of the Naṟṟiṇai + Glossary*. Critical Texts of Caṅkam Literature 1.1–1.3. 3 vols, Chennai: EFEO/Tamilmann Patippakam.

Paripāṭal.

Gros, François. 1968. *Le Paripāṭal. Texte Tamoul. Introduction, Traduction et Notes*. Publications de l'Institut Français d'Indologie 35. Pondichéry: Institut Français d'Indologie.

*Puṟanāṉūṟu*. [1996] 2002. *Puṟanāṉūṟu vol. 1*. Tinnelvelly: The South India Saiva Siddhanta Works Publishing Society LTD. First published 1996.

Viṣṇupurāṇa

*Viṣṇupurāṇa*. 1967. *Viṣṇupurāṇa and Śrīdhara's Commentary*. Bombay: Veṅkateśvara Press.

## Secondary References

Banerjee, Priyatosh. 1978. *The Life of Krishṇa in Indian Art*. New Delhi: National Museum.
Brereton, P. Joel, and Stephanie W. Jamison. 2020. *The Rig Veda, a Guide*. New York: Oxford University Press.

---

90  For the impact on the Muslim Indian world (Hawley [1983] 1989, pp. 11–12). J. S. Hawley has published several studies where he questions the literary, sociological, and political elaboration of the journey of the Bhakti movement said to have been travelling from south to north (Hawley 2015 for more). Regarding the circulation from north to south, and south to north (See Schmid 2014; Francis and Schmid 2014); where specificities of the contribution of the south are alluded to.

Brinkhaus, Horst. 2000. *The Mārkaṇḍeya-Episode in the Sanskrit Epics and Purāṇas. On the Understanding of Other Cultures. Proceedings of the International on Sanskrit and Related Studies to Commemorate the Centenary of the Birth of Stanislaw Schayer (1899–1941)*. Edited by Piotr Balcerowitz and Marek Mejor. Warsaw: Oriental Institute, Warsaw University, pp. 59–70.

Buck, David C., and K. Paramasivam. 1997. *The Study of Stolen Love: A translation of Kaḷaviyal eṉṟa Iraiyaṉār Akapporuḷ, with Commentary by Nakkīraṉ*. Georgia: Scholars Press of Atlanta.

Couture, André. 1991. *L'enfance de Krishna*. Paris and Laval: Le Cerf.

Couture, André. 1992. Le Bālacarita attribué à Bhāsa et les enfances hindoues et jaina de Kṛṣṇa. *Bulletin d'Études Indiennes* 10: 113–44.

Couture, André. 2007. *La Vision de Mārkaṇḍeya et la Manifestation du Lotus. Histoires Anciennes Tirées du Harivaṃśa (éd. cr., Appendice I, no 41)*. Paris: École Pratique des Hautes Études, Sciences historiques et philologiques.

Couture, André, and Christine Chojnacki. 2014. *Krishna et ses Métamorphoses dans les Traditions Indiennes, Récits D'enfance Autour du Harivamsha*. Paris: Presses Universitaires de la Sorbonne.

Couture, André, and Charlotte Schmid. 2001. The Harivaṃśa, the Goddess Ekānaṃśā and the Iconography of the Vṛṣṇi Triads. *Journal of the American Oriental Society* 121: 173–92. [CrossRef]

Cutler, Norman. 1987. *Songs of Experience. The Poetics of Tamil Devotion*. Bloomington: Indiana University Press.

Edholm, Erik Af, and Carl Suneson. 1972. The Seven Bulls and Kṛṣṇa's Marriage of Nīlā/Nappiṉṉai in Sanskrit and Tamil Literature. *Temenos: Studies in Comparative Religion* 8: 29–53. [CrossRef]

Francis, Emmanuel, and Charlotte Schmid. 2014. *Introduction: Towards an Archaeology of Bhakti. The Archaeology of Bhakti I. Mathurā and Maturai, Back and Forth*. Edited by Emmanuel Francis and Charlotte Schmid. Pondicherry: Institut Français de Pondichéry/École française d'Extrême-Orient, pp. 1–29.

Gail, A. J. 1969. *Bhakti im Bhagavatapurana*. Wiesbaden: Harrassowitz.

Griffith, Ralph T. H. 1973. *The Hymns of the Ṛgveda*. Delhi: Motilal Banarsidass.

*Prahlāda: Werden und Wandlungen einer Idealgestalt*. Wiesbaden: Akademie der Wissenschaften und der Literatur.

Hardy, Friedhelm. 1983. *Viraha-Bhakti, the Early History of Kṛṣṇa Devotion in South India*. Delhi: Oxford University Press.

Hart, George L., and Hank Heifetz. 2002. *The Puṟanāṉūṟu. Four Hundred Songs of War and Wisdom, an Anthology of Poems from Classical Tamil*. Penguin Books. New York: Columbia University Press. First published 1999.

Hawley, John Stratton. 1989. *Krishna, the Butter Thief*. Delhi: Oxford University Press.

Hawley, John Stratton. 1987. Krishna and the Birds. *Ars Orientalis* 17: 137–61.

Hawley, John Stratton. 2009. *The Memory of Love, Sūrdās Sings to Krishna*. New York: Oxford University Press, 2009.

Hawley, John Stratton. 2015. *A Storm of Songs: India and the Idea of the Bhakti Movement*. Cambridge: Harvard University Press.

Hudson, D. Dennis. 2002. Rādhā and Piṉṉai: Diverse manifestations of the same goddess. *Journal of Vaiṣṇava Studies* 10: 115–53.

Mahadevan, Iravathan. 2003. *Early Tamil Epigraphy. From the Earliest Times to the Sixth Century A.D.* Chennai and Cambridge: Cre-A and the Department of Sanskrit and Indian Studies, Harvard University.

Mysore Archaeological Department. 1938. *Mysore Archaeological Department, Annual Report-1936 (MAR 1936)*; Bangalore: Government Press, pp. 73–80.

Parthasarathy, R. 2004. *The Cilappatikāram of Iḷaṅkō Aṭikaḷ. An Epic of South India, Translated, with an Introduction and Postscript*. New York: Columbia University Press. First published 1993.

Podzeit, Utz. 1992. A philological reconstruction of the oldest Kṛṣṇa-epic. *Wiener Zeitschrift für die Kunde Südasiens und Archiv für indische Philosophie* 36: 55–59.

Preciado-Solis, Benjamin. 1984. *The Kṛṣṇa Cycle in the Purāṇas*. Delhi: Motilal Banarsidass.

Rajan, K. 2015. *Early Writing System. A Journey from Graffiti to Brāhmī*. Madurai: Pandya Nadu Centre for Historical Researches.

Ramanujan, Attipate Krishnaswamy. 1994. *Interior Landscapes, Love from a Classical Tamil Anthology*. Delhi: Oxford University Press. First published 1967.

Rocher, Ludo. 1986. *The Purāṇas*. Wiesbaden: Harrassowitz.

Schmid, Charlotte. 1997. Les Vaikuṇṭha Gupta de Mathurā, Viṣṇu ou Kṛṣṇa? *Arts Asiatiques* 52: 60–80. [CrossRef]

Schmid, Charlotte. 1999. Représentation anciennes de Kṛṣṇa luttant contre le cheval Keśin sur des haltères: l'avatāra de Viṣṇu et le dieu du Mahābhārata. *Bulletin de l'École française d'Extrême-Orient* 86: 65–104. [CrossRef]

Schmid, Charlotte. 2002. Aventures divines de Kṛṣṇa: la līlā et les traditions narratives des temples cōḻa. *Arts Asiatiques* 57: 33–50. [CrossRef]

Schmid, Charlotte. 2010. *Le Don de Voir, Premières Représentations Krishnaïtes de la Région de Mathurâ*. Paris: Publications de l'École française d'Extrême-Orient.

Schmid, Charlotte. 2013. The contribution of Tamil literature to the Kṛṣṇa figure of the Sanskrit texts: the case of the kaṉṟu in Cilappatikāram 17. In *Bilingualism and Cross-Cultural Fertilisation: Sanskrit and Tamil in Mediaeval India*. Edited by Whitney Cox and Vincenzo Vergiani. Pondichéry: École Française d'Extrême-Orient/Institut Français de Pondichéry, pp. 15–52.

Schmid, Charlotte. 2014. *Sur le Chemin de Krsna*. Paris: École française d'Extrême-Orient.

Schmid, Charlotte. forthcoming. Elements for an Iconography of Nārāyaṇa in the Tamil land: Balarāma and a lost Vaiṣṇava world. In *Viṣṇu-Nārāyaṇa: Changing Forms and the Becoming of a Deity in Indian Religious Tradition*. Edited by Marcus Schmücker. Vienna: Österreichische Akademie der Wissenschaften, phrase indicating stage of publication.

Takanobu, Takahashi. 1995. *Tamil Love Poetry and Poetics*. Leiden, New York and Köln: Brill.

Tieken, Herman J. H. 2001. *Kavya in South India. Old Tamil Cankam Poetry*. Groningen: Egbert Forsten.

Tieken, Herman J. H. 2003. Old Cankam literature and the so-called Cankam period. *The Indian Economic and Social History Review* 40: 247–78. [CrossRef]

Venkatesan, Archana. 2014. *A Hundred Measures of Time: Tiruviruttam by Nammalvar*. Translated and Introduced by Archana Venkatesan. London: Penguin Books.

