# Peer review of "The Carving of Kṛṣṇa’s Legend: North and South, Back and Forth"

_religions, doi:10.3390/rel11090439_

Round 1

Reviewer 1 Report

This is an excellent study, and I enjoyed reading it. The only section that could be slightly improved in terms of the text is the introduction, which almost feels double. This may well have been a requirement of the publisher, so in this case my comment can be ignored.

I am aware that the term "shadow-motif" has been taken from an earlier study, but do not feel it adequately represents the phenomenon described. It is somewhat negative, but what is happening is a positive embellishment, where a motif is taken to develop a new part of the story. This reminds very much of story telling and the ways a good story teller can embellish a story according to the time he has and the demands of his listeners. Another aspect that came to mind when reading through the work is the relationship of story telling to memorisation to explain some of the observed phenomena, as well as the use of painted textiles as the most likely candidate for transfer across regions. This would not be part of the often cited 'folk' tradition, but an expression of a profession. 

I am not sure if the child in Varanasi relief can simply be explained away as part of the village context, for that it is too prominent in the picture, but this does not invalidate the proposed hypothesis that the images are integral part of the tradition and transmission, a point that cannot be made often enough in a text centred world.

Minor issues I noticed:

p.2, l.71, double capitals

l.88, probably needs to be deleted.

Figure 1: beginning of caption missing.

l. 333-335: Is that paragraph part of the text?

l. 631: the insert in this translation is unclear.

l. 1072: is it is > as it is

l. 1121: I would rather speak of a 'visual tradition' here 

Author Response

Dear reviewer, I appreciate your careful guidance.

Reviewer 2 Report

In general, I found the article interesting and well argued. I have several suggestions beginning from the specific and minor to the more general and more significant.

The article requires more careful editing. For example, on p. 3, notes 2 and 3 are identical. And on p. 21, notes 44 and 45 are identical. The article includes many block quotes. Though this is generally appropriate for the content, there are problems. For example, line 670 is incorrectly indented as if it is a part of a block quote. And line 760 appears to introduce a block quote. Maybe that is the first sentence in line 761, but then the author(s) analysis continues after that sentence, and doesn’t belong in a block quote. Lines 333-335 appear to be a note by the author or authors to herself/himself/themselves. If I’m correct about that, the author or authors should be ashamed of herself/himself/themselves that it was left in.

Moving on to more substantial matters, in line 116 the article says that Surdas was blind. All the hagiographies do say this. However, in his 1984 Sur Das: Poet, Singer, Saint, John Stratton Hawley argues that all the references to his own blindness in the earliest poems by Surdas seem to be metaphorical, about his lack of spiritual insight (29). Hawley speculates that the later hagiographical tradition that Sur was blind from birth may have arisen from a mistaken reading of these poems as literal. However, admittedly this is not a central point in the argument of the article.

While we are on the subject of John Hawley, the article appropriately cites his work on Surdas and on the image of Krishna as a butter thief. However, the article does not refer to the recent (2015) and important book by Hawley, Storm of Songs: India and the Idea of the Bhakti Movement. The second chapter of that book is about a commonly held view that bhakti migrated from south India to north India, which Hawley problematizes. It would seem to me that this is quite germane to the argument of this article and it should be cited.

I take it that the central puzzle of the article is that although textual images of Krishna as a butter thief first emerge in Tamil poetry in south India, the earliest sculptural representations of this are from north India, and predate the texts. The eventual solution to this puzzle is that the early sculptures that appear to be of Krishna as a butter thief are not really about that at all, but are simply more or less generic representations of activities in a dairy farming village. More daringly and speculatively, the article suggests that the Tamil poets may have seen sculptures like these and have misread them as depictions of a specific story about the child Krishna, which they then proceeded to invent, without fully realizing that they were inventing something, since they thought they were following what they took to be an established story. The argument is interesting, and may be correct. The article does a good job of situating the history of the emergence of this motif in earlier Tamil literary images. A potentially serious problem with this argument for this reviewer is that the sculptural images of the butter thief who is not one that the article reproduces and analyzes are generally in a context where most of the images are of Krishna’s famous exploits. At one point (in line 308), the article describes some of its sculptures as “narrative relief friezes,” by which I think that it means that these are sculptures that you could use to tell a more or less continuous story of some of those exploits. That one of these images, the apparent butter thief one, is actually not of any well-known exploit, but is again a more or less generic depiction of life in a dairy farming village, is difficult to believe. Perhaps the article could bolster the case for this by showing other images that are a part of these sculptures that are not of Krishna’s exploits, if there are any.  

Author Response

Dear reviewer, I appreciate for your time devoted to religions-863956 and also for your professional opinion.

Round 2

Reviewer 2 Report

i appreciate the effort the author has made to respond to my earlier comments. I think the article is ready to publish.